# How a volcanic arc influences back-arc extension: insight from 2D numerical models

Duo Zhang[1] and J. Huw Davies[1]

[1]School of Earth and Environmental Sciences, Cardiff University, Cardiff, CF10 3AT, the UK

**Correspondence:** Duo Zhang (zhangd27@cardiff.ac.uk)

**Abstract.** Investigating plate tectonics through the lens of back-arc extension in subduction systems, this study introduces a 'hot region' on the overriding plate (OP) in 2D thermo-mechanical models, simulating the role of an arc. The models identified two extension locations on the OP: Extension at the Hot region (Mode EH) or Extension surprisingly at a Far-field location which is about 750 km from the trench (Mode EF). The study also found that extension could occur at the same far-field location without a hot region when the OP is young and thin, or the subducting plate (SP) is old with high sinking velocity. Our models suggest that the EH mode is common, occurring in many cases like Mariana Trough and Lau Basin, while the EF mode is rare, potentially occurring in scenarios like the Japan Sea. The primary driving mechanism in our models is poloidal flow beneath the OP, and the extension process is the competition of basal drag which thins the OP versus thermal healing which thickens it, and also a competition between thermal weakening at the hot region and at the far-field location. Increased trench retreat rates, facilitated by increased hot region temperature and width, encouraged this flow and consequently promoted back-arc extension.

## 1 Introduction

Back-arc extension is the first stage of the formation of a back-arc basin (BAB) which is found at the rear of an arc on the overriding plate (OP) in a subduction system. It provides a significant window into plate tectonics by displaying both convergent and divergent margins.

The driving mechanisms of back-arc extension have been widely explored by laboratory (Duarte et al., 2013; Clio and Pieter, 2013; Chen et al., 2016) and numerical modeling (Schellart and Moresi, 2013; Baitsch-Ghirardello et al., 2014; Holt et al., 2015; Sheng et al., 2019; Dal Zilio et al., 2018; Wolf and Huismans, 2019; Erdős et al., 2022). Numerous factors that influence back-arc extension have been investigated, including plate ages (Capitanio et al., 2011; Sheng et al., 2019; Dasgupta et al., 2021), slab width (Schellart and Moresi, 2013; Magni et al., 2014), slab dip (Lallemand et al., 2005; Dasgupta et al., 2021), and negative buoyancy of the slab (Chen et al., 2015). Many of the previous models concentrated on the effect of the subducting plate (SP) which has been identified as a driving force for mantle flow and plate tectonics (Schellart, 2004), while some other studies have explored the role that OP properties play in back-arc extension (Clio and Pieter, 2013). Moreover, the majority of the models listed above include a homogeneous OP, which eliminates the impact that possible weak regions (such as previous suture zones or volcanic arcs) may have on the results. There are a few numerical studies that introduce

inherited weak zones on the OP by lowering the viscosity directly (Nakakuki and Mura, 2013; Dal Zilio et al., 2018; Yang et al., 2019, 2021), but they primarily focus on simulating 'suture zones' on the continental lithosphere. There are also some works about volcanic arcs and back-arc extensions. For example, Corradino et al. (2022) and Baitsch-Ghirardello et al. (2014) have considered models including the arc formation process and its role in upper plate rifting. In these studies volcanic arcs formed self-consistently driven by the gradual hydration of the mantle wedge. Our work, in contrast, studies directly the effect of an instantaneous hot region on extension.

Even though BABs are all referred to as 'back'-arc basins, most of them were formed by breaking volcanic island arcs apart, leaving a remnant arc on the other side of the basin, such as Lau Basin which was formed by splitting Tonga Ridge and Lau Ridge apart (Zellmer and Taylor, 2001), Havre Trough between Kermadec Ridge and Colville Arc, Mariana Trough between Mariana Arc and West Mariana Ridge, etc. This phenomenon demonstrates that the presence of a volcanic arc can be important for constraining the location of extension, but few studies have taken this into account. To investigate this further, we have in this study run a series of 2-D thermo-mechanical and self-consistently driven models with a hot region that simulates a thermal volcanic arc on the OP. We will interchangeably use the terms volcanic arc / arc / hot zone knowing that readers understand that this is shorthand for approximating the effect of an arc with a hot zone. Firstly, we tested a series of models which have homogeneous OPs without arcs for comparison. Then, an arc was introduced into the OP to investigate its role in back-arc extension. We aim to look into how an arc influences back-arc extension and its position.

A remaining issue is that the properties of a volcanic arc are not very well known. Even though the location of present-day arcs is known from the Smithsonian Global Volcanism Project database, the arc-trench distances range widely from 2-1539 km data from Earthbyte, (Jodie, 2016). Additionally, the temperatures of arcs before their extension are difficult to constrain. Thus, we introduced a simplified model for the thermal signature of an arc, which allowed us to test a range of parameters for the arcs (including their width, arc-trench distance, and the central temperature anomaly) systematically to figure out their roles in back-arc extension.

## 2  Model description

Based on the work of Garel et al. (2014), a series of 2-D thermally-driven subduction models are run using the code Fluidity, a computational modeling framework suitable for geodynamic models, which uses an adaptive unstructured mesh that can be optimised dynamically (Davies et al., 2011). Fluidity provides a higher resolution in the region where the fields are changing most quickly spatially and a lower resolution in the regions where the field of the prognostic variable is stable and hardly changing. For example, in this 2D subduction model in a domain of 10,000 km by 2,900 km, adaptivity allows the model to have grid spacing ranging from 400 m near the subduction zone interface between the SP and OP, to 200 km in more quiescent regions in the deepest mantle. The grid adapts throughout the simulation, keeping the finest resolution where the spatial gradients of fields (viscosity, second invariant strain-rate, temperature and weak zone phase amount) are highest. It is a significant advantage in that it retains the accuracy of the solution while reducing the calculation effort.

## 2.1 Model setup

A 2D thermo-mechanical model of the subduction system was built with a juxtaposed subducting plate (SP) and overriding plate (OP) by Garel et al. (2014). Based on this model, we further tested the role that the initial plate ages near the trench play on back-arc extension and introduced a hot region on the OP.

The model built by Garel et al. (2014) has a large domain to reduce the influence of boundary conditions, which is 10000 km in length and 2900 km (the whole mantle depth) in depth (Figure 1), and a thin weak layer, with lower maximum composite viscosity and lower friction coefficient, lies on the SP and decouples the two plates to encourage the subduction of the SP. In the initial condition, the SP is set to bend beneath the OP with a radius of curvature of 250 km, and the depth of the tip of SP is 194 km, that is, the SP has already started subduction at the beginning of the model. In this way, the problem of subduction initiation is avoided. The two corners on the top surface represent ridges of the two plates, where the initial age is 0 Ma. The intersection of the plates is the position of the trench, where we set the initial age of plates as $\text{Age}^0_{SP}$ and $\text{Age}^0_{OP}$. The initial ages of plates have a linear change horizontally from 0 to $\text{Age}^0_{SP}$ and $\text{Age}^0_{OP}$. Velocity boundary conditions are free-slip on the bottom and the two sides, whereas the top boundary has a free surface (Kramer et al., 2012).

High heat flow has been observed in active volcanic arcs (Manga et al., 2012), so arcs can be expected to be described by a relatively high temperature. In the initial thermal structure, a hot region has been introduced on the OP in Garel's model to simulate the volcanic arc (Figure 1). The temperature at a chosen distance (from 100 to 1050 km) from the trench is increased by a certain degree (from 25 to 800 degrees) vertically from the mantle depth (which varies with the age of OP) to the surface. It cools off by a linear function over a prescribed horizontal distance (named 'Width', from 10 to 50 km) to merge in with the background temperature. The mantle temperature limits the maximum value, which is 1300°C. The hot region extends in depth from the surface to where it reaches the maximum temperature. The hot region has a higher temperature than the background, so its viscosity is lower as well (producing a pre-existing weak zone in the OP at the start of the simulation).

## 2.2 Governing equations

The governing equations of mantle dynamics follow the conservation law of mass, momentum, and energy. We solve the simplest set of equations by assuming an incompressible mantle and the Boussinesq approximation (McKenzie et al., 1974). The momentum and continuity equations are

$$\partial_i u_i = 0, \tag{1}$$

$$\partial_i \sigma_{ij} = -\Delta \rho g_j = \alpha \rho_s \left( T - T_s \right) g_j, \tag{2}$$

and the evolution of the thermal field follows:

$$\frac{\partial T}{\partial t} + u_i \partial_i T = \kappa \partial_i^2 T, \tag{3}$$

**Figure 1.** Model setup and initial geometry of the subduction simulations with the initial thermal structure of the hot region (modified from Garel et al. (2014)).

In these three equations, $\kappa$ denotes the thermal diffusivity, $u_i$ and $g_j$ are the vectors of velocity and gravity (which is oriented vertically downwards), respectively. In the $\Delta\rho$ (density anomaly term), $T$ is the temperature, $T_s$ is the temperature at the Earth's surface and $\rho_s$ is the nominal density at $T_s$, $\alpha$ is the coefficient of thermal expansion, and $\sigma_{ij}$ is the stress tensor which can be decomposed into deviatoric and non-deviatoric components, according to

$$\sigma_{ij} = \tau_{ij} - p\delta_{ij}, \tag{4}$$

with $\tau_{ij}$ as the deviatoric stress tensor, p is dynamic pressure, and $\delta_{ij}$ is the Kronecker delta function.

$\tau_{ij}$ the deviatoric stress tensor and $\dot{\varepsilon}_{ij}$, the strain rate tensor are related by $\mu$, the viscosity

$$\tau_{ij} = 2\mu\dot{\varepsilon}_{ij} = \mu\left(\frac{\partial u_i}{\partial x_j} + \frac{\partial u_j}{\partial x_i}\right), \tag{5}$$

## 2.3 Rheology

Four deformation processes of rocks that occur in the slab and mantle are considered: diffusion creep, dislocation creep, Peierls creep, and yielding. The first two mechanisms dominate the deformation at high temperatures, with diffusion creep being important in a low-stress environment, while dislocation creep is most significant under high stress. All of them are temperature-dependent except for yielding viscosity, and they have the following generic relationship with temperature, stress, activation energy, the prefactor, and strain rate:

$$\mu_{diff/disl/P} = A^{-\frac{1}{n}} exp\left(\frac{E+PV}{nRT_r}\right)\dot{\varepsilon}_{II}^{\frac{1-n}{n}}, \tag{6}$$

In this equation, A is a prefactor, n is the stress exponent with a value of 1, 3.5 and 20 for diffusion, dislocation and Peierls creep, respectively; E is the activation energy, which is 300 kJ/mol in diffusion creep (200 kJ/mol in the lower mantle), 540 kJ/mol in dislocation creep and 540 kJ/mol in Peierls creep of the upper mantle (Hirth and Kohlstedt, 2003; Karato and Wu, 1993; Karato, 1997); P is the lithostatic pressure, given by $P=\rho_s gz$, $\rho_s$ is the reference density at the Earth's surface temperature $T_s$, g is the acceleration due to gravity and z is the depth; V is the activation volume, R is the gas constant, and $\dot{\varepsilon}_{II}$ is the second invariant of the strain rate tensor. Temperature (T), on which the rheology of slabs and mantle most depend, is given initially as

$$T\left(x,z,t=0\right) = T_s + \left(T_m - T_s\right)erf\left(\frac{z}{2\sqrt{\kappa Age^0\left(x\right)}}\right),\tag{7}$$

where z is the depth, x is the horizontal coordinate, and t is the time, $T_s$ the temperature at Earth's surface and $T_m$ the mantle temperature. $T_r$ in Eq. 6 is the temperature in Eq. 7 adding an adiabatic gradient of 0.5 K/km in the upper mantle and 0.3 K/km in the lower mantle (Fowler, 1990).

In addition, the yielding viscosity is given by

$$\mu_y = \frac{\tau_y}{2\dot{\varepsilon}_{II}},\tag{8}$$

with $\tau_y$ the yield strength:

$$\tau_y = min\left(\tau_0 + f_c P, \tau_{y,max}\right),\tag{9}$$

$\tau_0$ is the surface yield strength (2 MPa), $f_c$ is the friction coefficient (0.2), P is the lithostatic pressure, and $\tau_{y,max}$ is the maximum yield strength (10,000 MPa). The friction coefficient of 0.2 in our models is intermediate between lower values of previous subduction models (Di Giuseppe et al., 2008; Crameri et al., 2012).

For Peierls creep, we use the Eq. 6 to follow the simplification in Čížková et al. (2002). A high exponent n (20 in our work) approximates the exponential strain rate and the temperature dependency of the Peierls mechanism.

The composite viscosity is based on the combination of these mechanisms via

$$\mu = \left(\frac{1}{\mu_{diff}} + \frac{1}{\mu_{disl}} + \frac{1}{\mu_y} + \frac{1}{\mu_P}\right)^{-1}.\tag{10}$$

This is assuming that the strain rates of all 4 deformation processes sum, as is the case for viscous dashpots in series (Schmeling et al., 2008). There are 2 materials set in our models, and the whole domain is controlled by similar rheological laws in both of them. The second material is in a weak (5 km thick) layer upon the subducting plate tracked down to 194 km depth, and graded out below 200 km depth. It is the same parameters as the other material covering virtually the whole domain, other than it has a very low friction coefficient (one-tenth of that in the normal material), and a lower maximum viscosity

 of $10^{20}$ Pa s ($10^{25}$ Pa s in normal material) to ensure the decoupling between the subducting and overriding plates. All the parameters of the model setup are listed in Table 1.

Table 1: Key parameters used in this research.

| Quantity | Symbol | Units | Value |
|---|---|---|---|
| Gravity | $g$ | $\mathrm{m\,s^{-2}}$ | 9.8 |
| Thermal expansivity coefficient | $\alpha$ | $\mathrm{K^{-1}}$ | $3\times10^{-5}$ |
| Thermal diffusivity | $\kappa$ | $\mathrm{m^2\,s^{-1}}$ | $10^{-6}$ |
| Reference density | $\rho_s$ | $\mathrm{kg\,m^{-3}}$ | 3300 |
| Cold, surface temperature | $T_s$ | K | 273 |
| Hot, mantle temperature | $T_m$ | K | 1573 |
| mantle geothermal gradient | $G$ | $\mathrm{K\,km^{-1}}$ | 0.5 (UM) |
| | | | 0.3 (LM) |
| Gas constant | $R$ | $\mathrm{J\,K^{-1}\,mol^{-1}}$ | 8.3145 |
| Maximum viscosity | $\mu_{max}$ | Pa s | $10^{25}$ |
| Minimum viscosity | $\mu_{min}$ | Pa s | $10^{18}$ |
| **Diffusion Creep** | | | |
| Activation energy | $E$ | $\mathrm{kJ\,mol^{-1}}$ | 300 (UM) |
| | | | 200 (LM) |
| Activation volume | $V$ | $\mathrm{cm^3\,mol^{-1}}$ | 4 (UM) |
| | | | 1.5 (LM) |
| Prefactor | $A$ | $\mathrm{Pa^{-n}\,s^{-1}}$ | $3.0\times10^{-11}$ (UM) |
| | | | $6.0\times10^{-17}$ |
| | $n$ | | 1 |
| **Dislocation Creep (UM)** | | | |
| Activation energy | $E$ | $\mathrm{kJ\,mol^{-1}}$ | 540 |
| Activation volume | $V$ | $\mathrm{cm^3\,mol^{-1}}$ | 12 |
| Prefactor | $A$ | $\mathrm{Pa^{-n}\,s^{-1}}$ | $5.0\times10^{-16}$ |
| | $n$ | | 3.5 |
| **Peierls Creep (UM)** | | | |
| Activation energy | $E$ | $\mathrm{kJ\,mol^{-1}}$ | 540 |
| Activation volume | $V$ | $\mathrm{cm^3\,mol^{-1}}$ | 10 |
| Prefactor | $A$ | $\mathrm{Pa^{-n}\,s^{-1}}$ | $10^{-150}$ |
| | $n$ | | 20 |

| Yield Strength Law | | | |
|---|:---:|:---:|:---:|
| Surface yield strength | $\tau_0$ | MPa | 2 |
| Friction coefficient | $f_c$ | | 0.2 |
| | $f_{c,weak}$ | | 0.02 (weak layer) |
| Maximum yield strength | $\tau_{y,max}$ | MPa | 10,000 |

## 3 Results

Two sets of models with a hot region were run to test the role of properties of a hot region (Table A1) and plate ages (Table A2), respectively. Before that, we run models that only vary initial plate ages for comparison with those that contain a hot region on the OP (Table A2). The naming of the models depends on the parameter types and is shown in Table A1 and A2.

Model SP90_OP20 ($Age^0_{SP}$ is 90 Ma and $Age^0_{OP}$ is 20 Ma) without a hot region was chosen as a reference model (RM, Figure 2) for testing properties of the hot region. The subducting plate (SP) sinks rapidly and reaches the boundary between the upper mantle (UM) and lower mantle (LM) at about 4 Ma after model initiation. We note that the SP sinking velocity in this short time period increases to a very high value (with a local peak of about 38 cm/yr, and a lower peak when averaged over 1 Myr [geologically observable timeframe], as expected, as the length of the driving subducting slab increases, before decreasing to nearly steady values of < 2 cm/yr once the slab approaches and enters the more viscous lower mantle. During the time of rapid sinking, the OP basement near the trench is thinned by the mantle wedge flow, and the region that is about 750 km away from the trench on the OP becomes thinner, a region which is under extensional stress (Figure 2d). After the SP tip reaches the LM which is more viscous compared to the UM, its subduction slows down due to the increased resistance, which leads to the subduction process becoming steady state (Capitanio et al., 2007). The thinning regions of the OP lithosphere heal gradually during the steady state subduction stage, after which time the whole OP is under a compressional environment. The trench retreats all the time though the rate of retreat varies. The models that tested the effect of the properties of the hot region are all set the same $Age^0_{SP}$ and $Age^0_{OP}$ as the RM (Table A1).

### 3.1 Back-arc extension modes

The extension levels on the OP are defined by the isothermal contour of 1300 K (Figure 3). The contour curving up means the OP is thinned in thermal structure, which is defined as 'Thermal Thinning' (Figure 3b). When the contour curve reaches the cold thermal surface, the OP finishes thinning (Complete Thermal Thinning, Figure 3c) and turns to spread (Figure 3d) which is named 'Spreading'. The stage when the thermal thinning is completed is the critical point between Thermal Thinning and Spreading and is regarded as the Incipient Extension, after that the extensional state begins. Thus, we introduce two new classifications 'Extension' and 'No Extension' identified by capitalisation, where we classify Complete Thinning and Spreading as 'Extension' and the states before Complete Thinning as 'No Extension'.

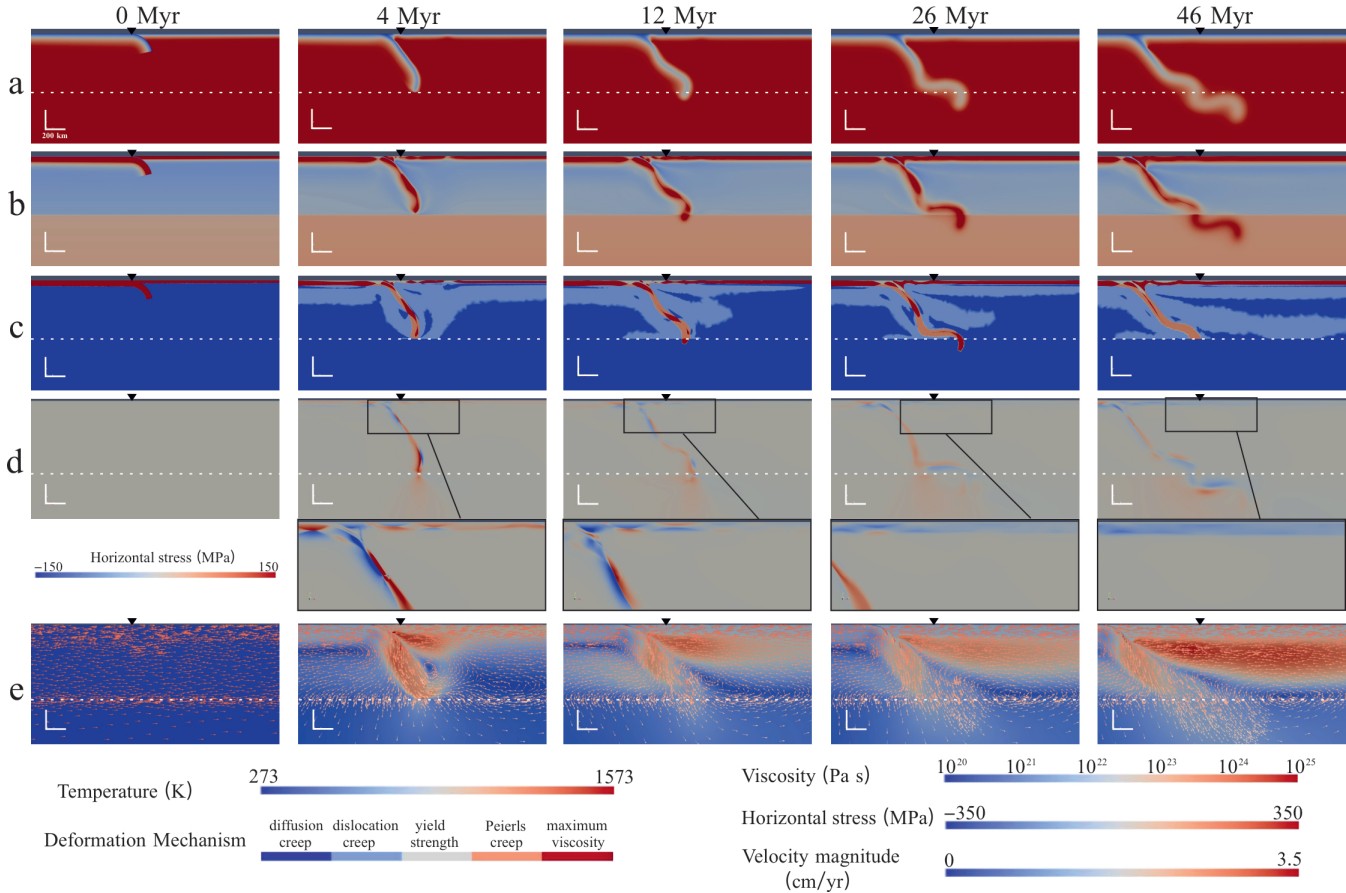

**Figure 2.** Simultaneous snapshots of a zoom in of the reference model showing the time evolution of (a) temperature field; (b) viscosity field; and (c) dominant deformation mechanism; (d) horizontal stress (positive value indicates extensional stress, and zoom-in stress field under each snapshot is shown for a clearer view, which uses a different colour bar from the original ones); and (e) the magnitude of velocity field with arrows, and the arrows only indicate directions while the colours represent the value. The black lower triangles mark the initial trench positions, and the dashed line represents the boundary of UM and LM. The horizontal and the vertical scales are the same, which are 200 km.

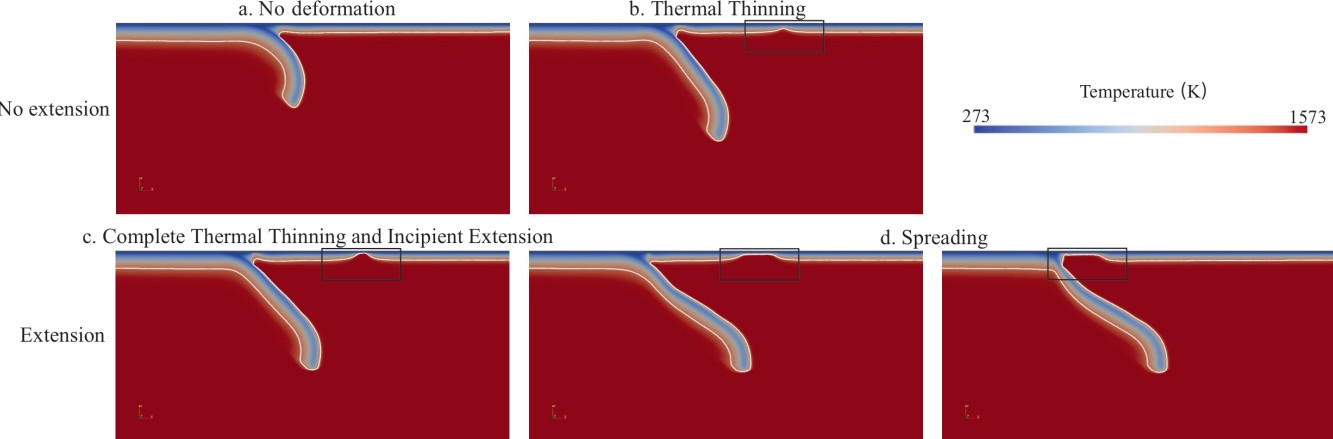

**Figure 3.** Definition of the extension levels by the isothermal contour of 1300 K (the white line). The temperature fields show (a) No deformation, (b) Thermal Thinning, (c) Complete Thermal Thinning and Incipient Extension and (d) Spreading (far-field spreading on the left and hot-region spreading on the right). (a) and (b) are classified as 'No Extension', whereas (c) and (d) are classified as 'Extension'.

Based on the above definition, three end-member modes of back-arc extension on the OP are recognised (Figure 4): (A) No Extension on the OP (NE), which means there is no actual spreading on the OP; (B) Extension at the Far-field location (around 750 km from the trench) which is far away from the hot region and the very same region where the thinning takes place in the RM (EF); (C) Extension at the Hot region (EH, EH-D). We also get the fourth mode (D) which has two (dual) extensions (Figure 5, i.e. at both places in modes B and C (EF+EH). The ones showing slab detachment after the back-arc extension have been subdivided into EH-D (Figure 5a, b), for the case in which the slab breaks off, in these cases, the plate loses its slab pull and both the subduction and trench retreat stops. The extension of all modes lasts for a short time (just 1 to several million years) and then heals gradually in the steady subduction state or after slab detachment (in EH-D mode).

The reason why we combine EH and EH-D into the same mode is that the slab detachment occurs after the back-arc extension and has no effect on driving the extension. The specific morphology is not the primary focus of this research.

### 3.2 Varying parameters of the hot region

More than 200 models have been performed by varying the following parameters of the hot region: the distance from the trench to the centre of the hot region (Distance), the width from the heated centre of the hot region to each side (Width), and the increased temperature at the centre of the hot region from the background temperature ($\Delta$T)(shown in Figure 1 and Table A1), to investigate the influence of a pre-existing hot region (a volcanic arc) on back-arc extension.

#### 3.2.1 Regime diagrams

In this section, we mainly show how the modes change through changes in the above parameters (Figure 6, 8).

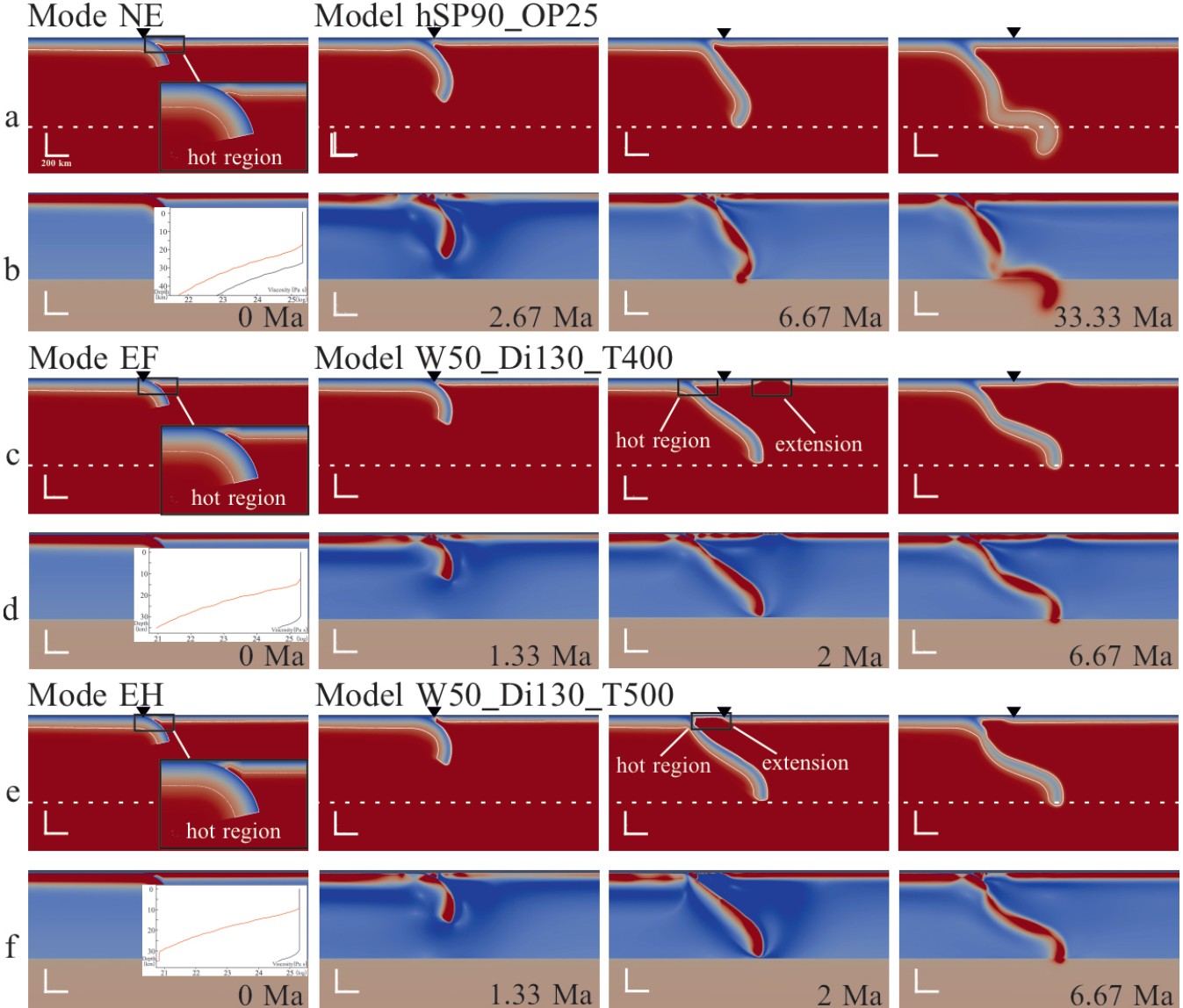

**Figure 4.** Three end-member modes of the back-arc extension on the OP: (a)(b) NE; (c)(d) EF; (e-h) EH. Figures (a) (c) (e)show the temperature field with the isothermal contour of 1300 K in white, and we have a zoom-in showing the hot region in the 0 Ma figure of each model. Figures (b) (d) (f) show the viscosity field and plots of the initial vertical viscosity at the hot region centre in the 0 Ma figure of each model represented by the red lines, while the black lines represent the viscosity without the hot region. The two lines in different colours show the change in viscosity resulting from emplacing a hot zone. The parameters of each model are shown in Table A1 and A2. In Model hSP90_OP25, $Age_{SP}^0$ is 90 Ma and $Age_{OP}^0$ is 25 Ma, 'h' means a hot region is included and the properties are: Width = 50 km, Distance = 150 km, $\Delta T$ = 100. W50_Di130_T400-500 means that Width is 50 km, Distance is 130 km, $\Delta T$ is 400 and 500°C, respectively, and their plate ages are the same as the reference model ($Age_{SP}^0$ is 90 Ma and $Age_{OP}^0$ is 20 Ma)

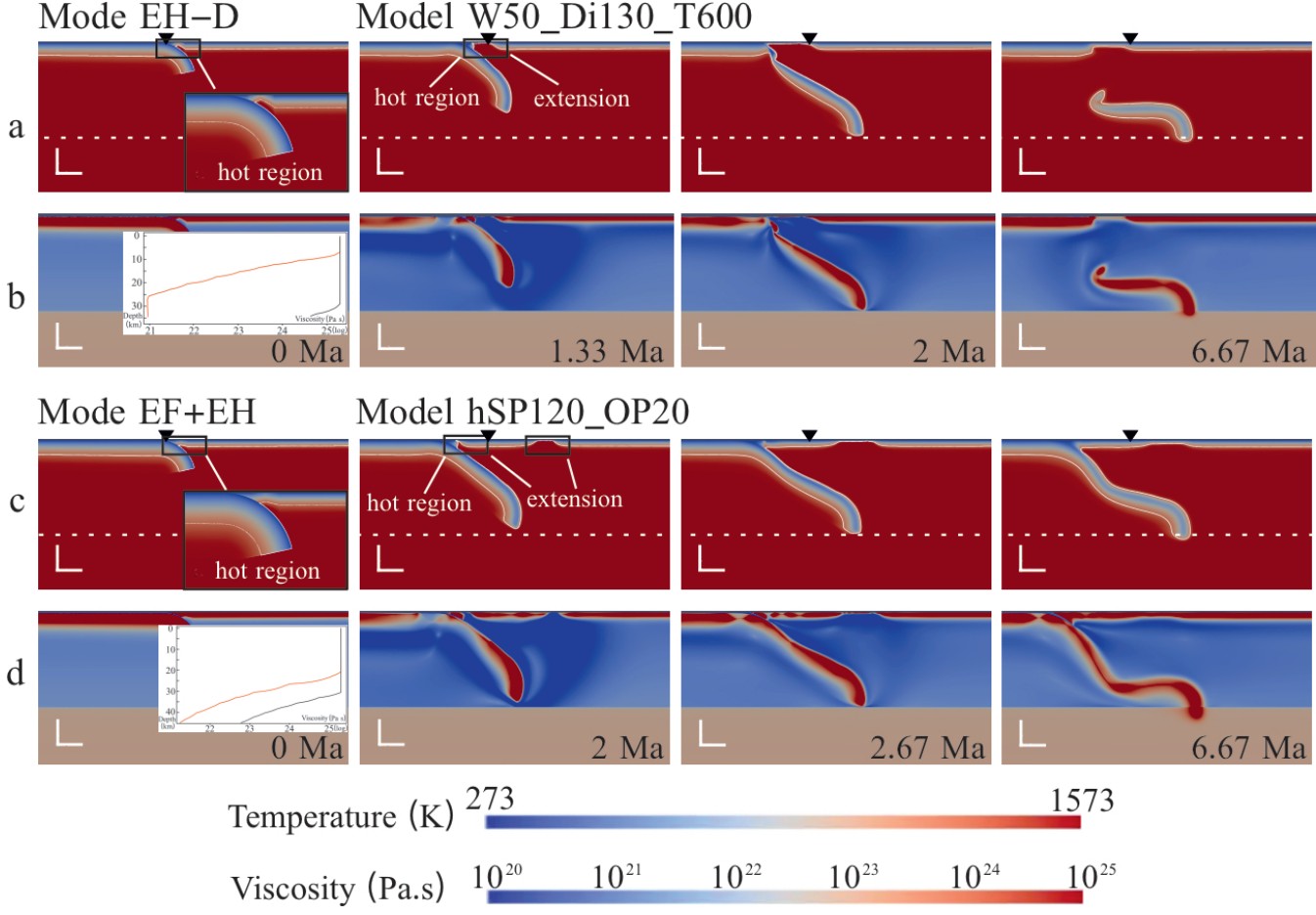

**Figure 5.** Supplementary modes of the back-arc extension: (a)(b) EH-D, which shows slab detachment; (c)(d) EF+EH. The same as that in Figure 4, Figures (a) (c) show the temperature field and Figures (b) (d) show the viscosity field and plots of the initial vertical viscosity at the hot region centre. The specific description is in Figure 4. Model hSP120_OP20 has a hot region with the same properties as hSP90_OP25 and different plate ages of 120 Ma and 20 Ma, respectively. W50_Di130_T600 means that Width is 50 km, the Distance is 130 km, $\Delta$T is 600°C, and its plate ages are the same as the reference model.

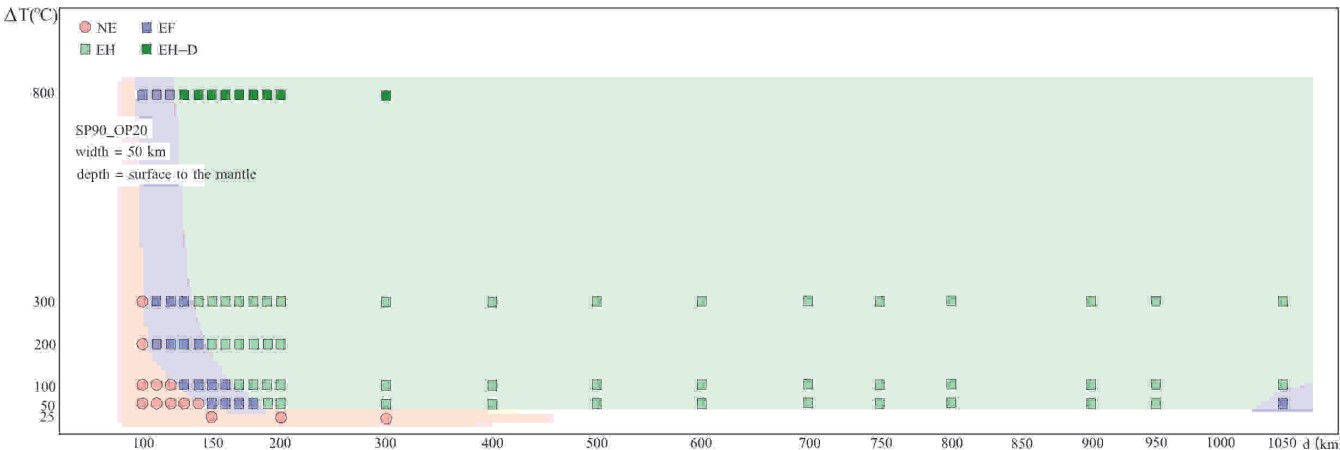

**Figure 6.** Regime diagram showing the Extension behaviour of the overriding plate with a hot zone. It is a function of the distance from the trench to the centre of the hot region (d, km) and the temperature anomaly of the hot region ($\Delta T$). The width of the hot region is fixed at 50 km.

Firstly, the Width was fixed to 50 km and the Distance and $\Delta T$ were tested systematically (Figure 6). All of the modes we introduced in section 3.1 are identified in this series of models. When the hot region is quite close to the trench, such as when the Distance is 100 km, the model behaves as mode NE (No Extension) no matter how hot the hot region is. As the hot region is emplaced farther away from the trench, an extension emerges at the far-field location (mode EF). When the $\Delta T$ is prescribed to 50 degrees, Extension cannot appear if the distance is less than 150 km. Whereas when the thermal anomaly is increased to 100 degrees, the Extension occurs even when the distance is only 130 km. In other words, the distance threshold controlling the emergence of Extension on the OP reduces as the temperature of the thermal anomaly increases.

As the distance increases further, the mode of the cases changes from EF to EH, which has Extension at the hot region close to the trench. The threshold of distance, controlling the transition from mode EF to EH, has the same trend as that controlling the existence of Extension. When the distance is larger than the threshold, most of the cases show mode EH as long as there is an Extension. However, model W50_Di1050_T50 (W50 means that the Width is 50 km; Di1050 represents the Distance is 1050 km; T50 means that the $\Delta T$ is 50 degrees) shows mode EF unexpectedly (Figure 7). In this model, the hot region is located at 1000-1100 km from the trench, but the Extension is still around 750 km from the trench. When the hot region is very close or very far away from the trench, the Extension could emerge near a distance of around 750 km from the trench.

To find out the role of the Width, we reduce it from 50 km to 25 km. A series of cases investigate how the combination of the Distance and $\Delta T$ changes the extension behaviour of the different-width hot region (Figure 8). The depth extent of the thermal anomaly has not been changed in these models. Compared to Figure 8a, the results in Figure 8b show that the thresholds between mode NE and EF, as well as mode EF to EH in distance and heating temperature both increase obviously. Take models with 100 degrees temperature increase for example, when the distance is 150 km, the model with a width of 50 km shows EF mode which has an Extension on the OP, but the model with a width of 25 km is in NE mode which does not

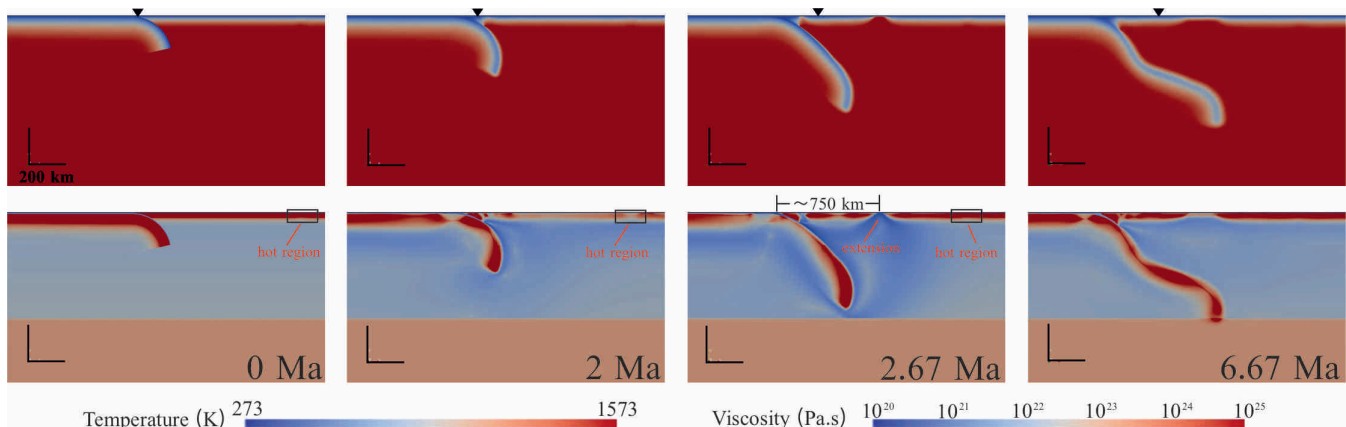

**Figure 7.** W50_Di1050_T50 in EF mode.

show Extension on the OP. Regardless of the specific values, the tendency of the changes in threshold when the width is 25 km
is the same as that with a width of 50 km. In summary, as one might expect, a wider hot region favours Extension.

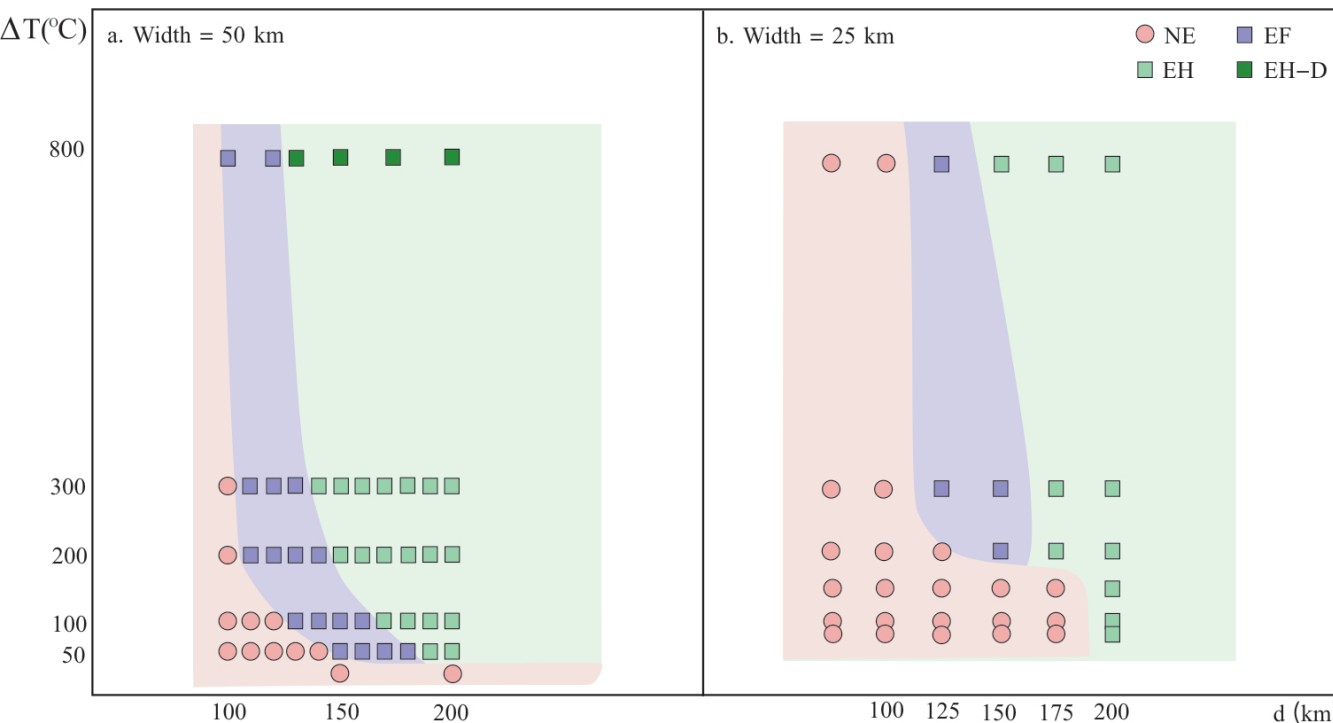

**Figure 8.** (a) Regime diagram of distance vs. $\Delta$T when the width is 50 km and (b) the comparison to the results when the width is 25 km.

In conclusion, a farther distance from the heated centre to the trench, a higher temperature, and a larger size of the hot region all favour back-arc extension on the OP.

### 3.2.2 Correlation to the trench retreat rate

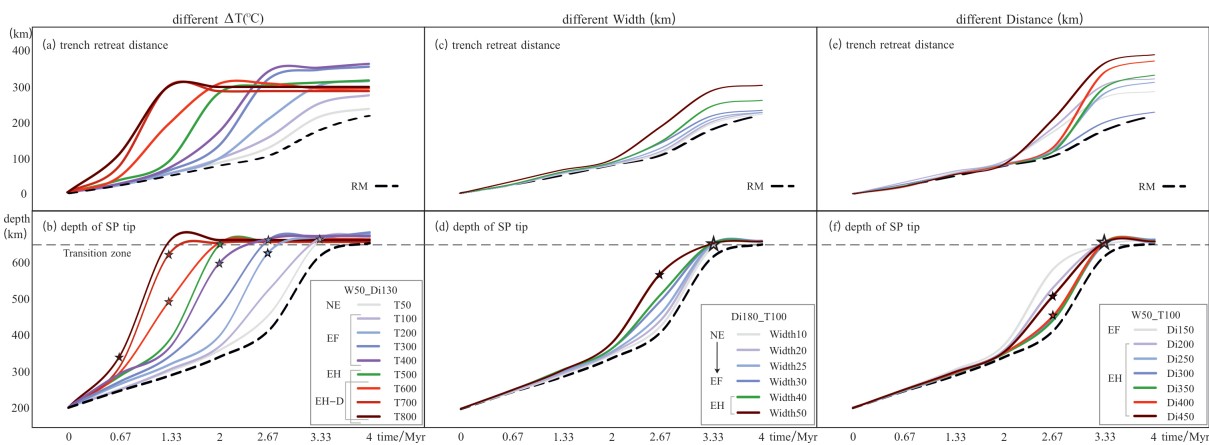

**Figure 9.** Comparisons of total (a) (c) (e) trench retreat and (b) (d) (f) depth of SP tip subduction throughout the first 4 Myr simulation with various ΔT (a, b), different Width (c, d) and various Distance (e, f), respectively. The grey dashed line represents the transition zone depth (660 km). The stars mark the time when the Extension occurs in each model, and the colour of the stars is the same as the lines of each model. The reference model (RM) is marked as the black dashed line.

The vertical series of models with the Distance value of 130 km in Figure 6 are chosen to investigate the influence of the ΔT. The ΔT varies from 50 to 800 degrees, and the modes that they show transfer from NE to EF, then to EH as the ΔT increases.

As stated in the description of the Reference Model (RM), the SP sinks rapidly initially, accelerating until the tip reaches the transition zone between the UM and the LM. After introducing a hot region into the OP, the SP sinks more rapidly on average and the trench retreats slightly faster (peaking from 10 to 23 cm/yr) before the SP reaches the lower mantle than in the RM. When the ΔT increases, the general SP sinking rate also rises, and the SP tip takes less time to get to the transition zone (Figure 9b). Like the vertical slab sinking rate, the trench retreat rate shows a similar trend as ΔT increases (Figure 9a). When the ΔT is no more than 400 degrees, the trench retreats at a similar speed at first, then the model with a hotter 'arc' shows a quicker trench retreat, after 3.33 Myrs, the trench keeps quite a slow retreating rate in the steady state stage. As the ΔT is increased further, the trench obviously retreats faster before the SP reaches the transition zone. The trench stops retreating after the slab detachment in Mode EH-D, because there is no extra force exerted on the SP.

Similarly, when the Width rises, both the SP rate and trench retreat rate slightly increase with the same trend as when ΔT increases (Figure 9c, d), though the increase with Width is much less than with ΔT. However, there is no specific correlation between Distance and trench retreat rate (Figure 9e) / SP sinking rate (Figure 9f). For example, compared to model

W50_T100_Di300 showing the slowest trench retreat rate except for the RM, the trench in model W50_T100_Di200 retreated more quickly and farther until 4 Myrs.

Whether increasing the Width or $\Delta T$, the hot region is hotter and weaker in total, which encourages the trench retreat by providing less horizontal resistance to the trench rollback. Whereas varying the Distance alone has no influence on the trench retreat since the heated area and temperature are unchanging. Besides, the EH-D did not occur in the two latter series of models, which possibly means that the $\Delta T$ has the most significant influence on the behaviours of the plates.

### 3.3   Varying initial plate ages adjacent to the trench

We chose the hot region located 100-200 km from the trench and heated it by 200 degrees (the same as that in Model W100_Di150_T200), and tested how different combinations of plate ages influence the OP deformation (Table A2). The results of changing the initial OP age and SP age at the trench, showing the modes of the OP deformation, are shown in Table A2 and Figure 10. In general, the figure reveals that older SP age tends to result in Extension when the OP age remains unchanged, whereas a younger OP age leads to Extension when the SP age is fixed.

In models without a hot region (modes changing shown by the dashed line), when the initial age of the OP adjacent to the trench ($\text{Age}_{OP}^0$) is prescribed to be 20 Ma, the threshold of the initial age of the SP at the trench ($\text{Age}_{SP}^0$) that leads to Extension is 100 Ma (Model SP100_OP20), and a little bit of thinning occurs at the place where Extension exists in Model SP100_OP20 if the $\text{Age}_{SP}^0$ is decreased to 90 Ma (Model SP90_OP20). We change the $\text{Age}_{OP}^0$ and find the minimum $\text{Age}_{SP}^0$ to generate Extension for each $\text{Age}_{OP}^0$. As seen in Figure 10, model SP60_OP15, SP95_OP20, SP165_OP25, and SP280_OP30 are the

marginal cases in the region in which the models have Extension. As $\text{Age}_{SP}^0$ increases from 70 Ma to 280 Ma, the time before Extension occurs gets slightly longer, from 2.7 Ma to 4.7 Ma. For these three models, increasing $\text{Age}_{OP}^0$ or decreasing $\text{Age}_{SP}^0$ stops Extension. All of this extension or thinning happens at a far-field location about 700-750 km from the trench.

    Adding a hot region increases the $\text{Age}_{OP}^0$ threshold of the existence of Extension, which is shown by the boundary of the pink area and the dashed line (Figure 10), and allows Extension to occur at the hot region which is much closer to the trench

than the far-field location. When the $\text{Age}_{SP}^0$ is relatively young, Extension is more likely to happen at the arc, though some cases showed far-field Extension before adding the hot region (eg.SP80_OP15 and hSP80_OP15, 'h' in the model name means a hot region is set in the model). Most models with an $\text{Age}_{SP}^0$ older than 100 Ma tend to lead to an EF mode. Furthermore, the transition modes in Figure 10 show how the end-member modes change when the initial plate ages near the trench vary. Take the OP20 series of models for example, when the $\text{Age}_{SP}^0$ increases from a low value (such as hSP60_OP20 which is mode

NE), Extension first occurs at the hot region (eg.hSP70_OP20; mode EH) and at the far-field location (mode EF) when $\text{Age}_{SP}^0$ is old enough (eg.hSP150_OP20). Transitional mode EH+EF is shown in this process as well.

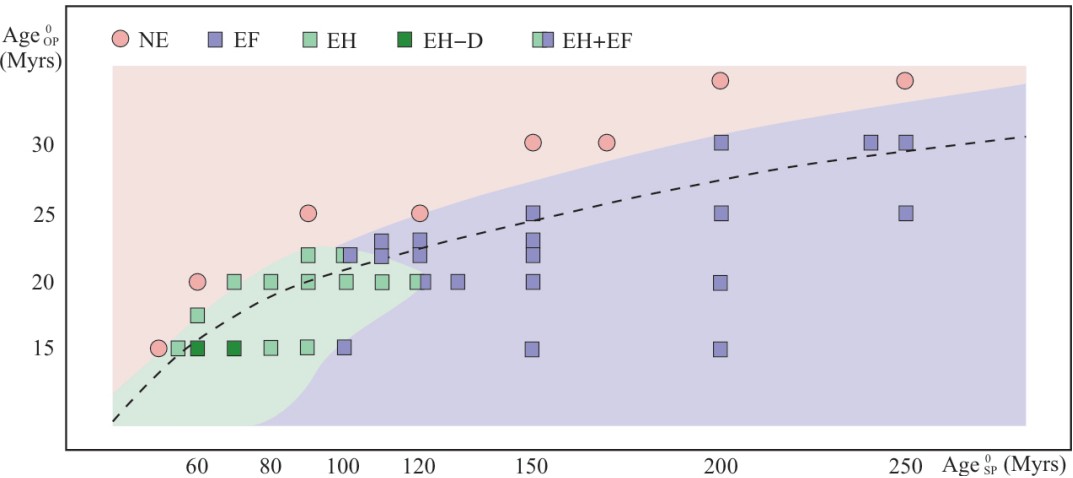

**Figure 10.** Regime diagram of the combination of $\text{Age}_{OP}^0$ and $\text{Age}_{SP}^0$. Each symbol represents a model with a hot region located from 100 to 200 km from the trench, and the different colour regions are classified by the results of models with a hot region. For comparison, the threshold for the existence of extension in models without a hot region is marked by the black dashed line. Note that one symbol with two squares means two end-member modes both happen in one model.

## 4 Discussion

### 4.1 The driving mechanism of the back-arc extension

Generally speaking, back-arc extension can be driven by extensional stresses related to plate motions (applied at the trench
boundary of the OP) and/or subduction-induced mantle flow (applied at the base of the OP). In the first mechanism the extensional stress is generated from the speed difference between the trench and the OP, while the second arises from variable shear stresses at the base of the OP. The plate will start to rift when the extensional stress exceeds the natural strength of the plate, which could result from increasing stress or decreasing strength. We further investigate these aspects to find which are most important in our models.

We calculated the vertically average horizontal stress ($\sigma_h$) over the OP thickness and plotted the maximum value ($\sigma_{hmax}$) for each simulation as a function of $\text{Age}_{SP}^0$ (Figure 11) in models without a hot region. The horizontal location for the stress calculation is the location of maximum $\sigma_h$ at the time of maximum $\sigma_h$. When the $\text{Age}_{SP}^0$ is relatively young, the $\sigma_{hmax}$ increases quickly as the $\text{Age}_{SP}^0$ rises from 50 to about 110 Myrs ($\text{Age}_{OP}^0$=30 Myrs). After this age, $\sigma_{hmax}$ no longer changes quickly as the $\text{Age}_{SP}^0$ keeps rising, even in cases when the $\text{Age}_{SP}^0$ is greater than 270 Myrs and in which Extension occurs
(Figure 11c). Most importantly, the $\sigma_{hmax}$ does not change when a hot region is introduced into the OP (marked by stars). In these models, the extensional force (indicated by $\sigma_{hmax}$) does not play a controlling role in back-arc extension, which implies the weakening of the OP dominates the process in these cases.

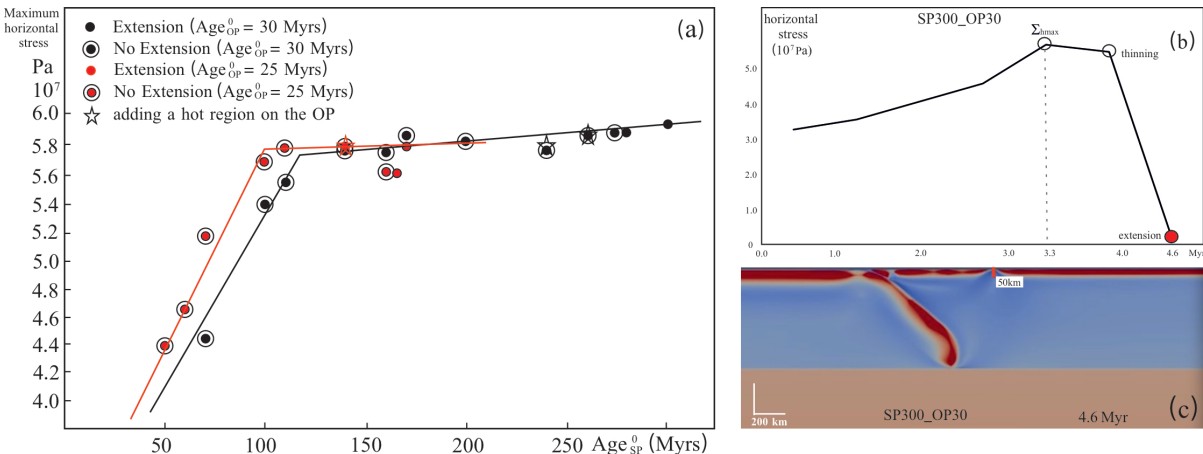

**Figure 11.** (a) Maximum vertically averaged horizontal stress ($\sigma_{hmax}$) over OP thickness (from the surface to the OP bottom, 50 km depth when the $Age_{OP}^0$ is 30 Myrs and 45 km depth when the $Age_{OP}^0$ is 25 Myrs which was defined by the temperature contour of 1300K.) vs. $Age_{SP}^0$. The stars mark the models which have a hot region on the OP. (b) Depicts the time evolution of vertically averaged horizontal stress ($\sigma_h$) and (c) identifies the specific location (in the viscosity field) where the stress in panel (b) is calculated. (b) and (c) are both from model SP300_OP30. A positive stress value means extensional.

The analysis of the horizontal stress shows a compressional region close to the trench just before the Extension occurs in all the models (Figure 12), which is consistent with that in RM (Figure 2d) and some previous numerical models (Chen et al.,
2016; Schellart and Moresi, 2013; Corradino et al., 2022; Balázs et al., 2022). Even though Extension in Mode EH occurs near the trench (orange star in Figure 12), the vertically integrated horizontal stress ($\Sigma_h$) shows no observable difference from that in EF mode. The introduction of the hot region, again as in Figure 11, has almost no influence on $\sigma_h$. Therefore, the change of $\Sigma_h$ or $\sigma_h$ does not really decide if Extension occurs, which suggests that the horizontal extensional stress (while required) is not the critical cause of Extension in our models. It is worth noting that these models have a mobile OP, facilitated by a ridge
on the right-hand boundary, and the conclusion might be different for fixed OP.

The RM shows only very limited thinning at approximately 750 km away from the trench on the OP (Figure 2). To produce back-arc extension, either the OP needs to weaken or the driving force needs to increase. In this series of models, the OP can be weakened in two ways. One is through the hot zone which was introduced in the initial setup, because viscosity decreases as temperature rises (Eq. 6), the hot zone would be easier to extend; the other way it could be weakened is by upwelling flow
beneath the OP. This latter process must have occurred in this series of models, as implied by the existence of the EF mode. In total, the main driving mechanism is poloidal flow induced by the subducting slab and shown by the velocity field (Figure 13). The trench-ward horizontal flow (X component) produces basal drag by the velocity gradient (Figure 14), which facilitates an Extension by producing the extensional force and weakening the OP, whereas the Y component of the flow also encourages the thermal weakening of the OP. The upwelling flow, the cause of both components of flow, is always in a similar direction
(about $60°$ to the horizontal) as it approaches the OP. The horizontal size of the upwelling flow cell varies a little, is around

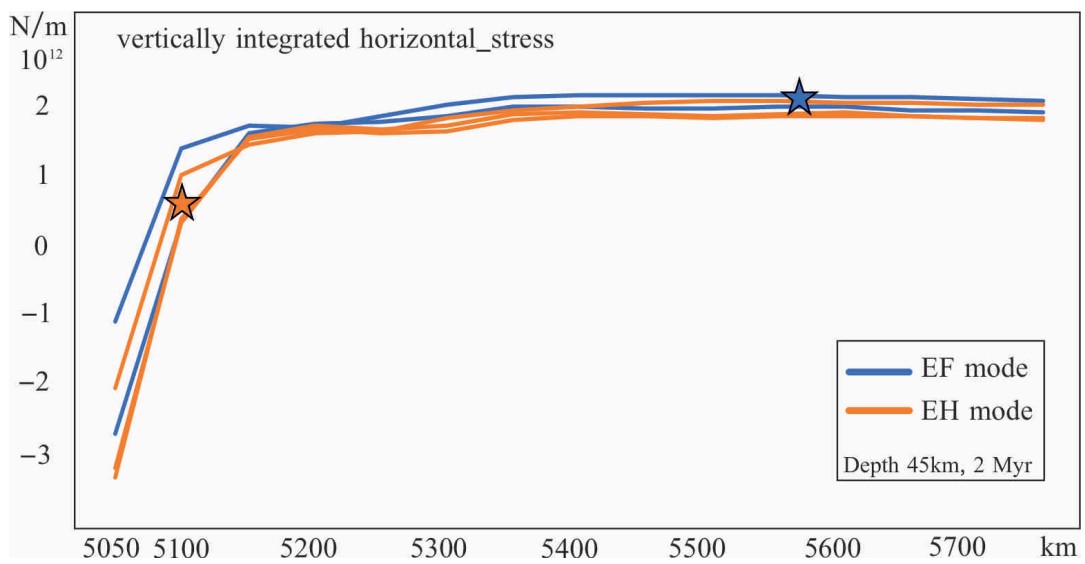

**Figure 12.** Vertically integrated horizontal stress over OP thickness (from the surface to the OP bottom, 45 km depth, which was defined by the temperature contour of 1300K) vs. x coordinate (the trench starts with an x coordinate of 5000 km). A positive stress value means extensional. For clarity, only two colours are used in this figure to represent models in different modes (see legend). The chosen models are W50_Di120_T100 and W50_Di150_T100 which are in EF mode; W50_Di170_T100, W50_Di190_T100, and W50_Di250_T100 which are in EH mode. Stars mark locations of back-arc extension in the same colour as their plots (eg. blue star represents the extensional location in EF mode), respectively. This is evaluated after 2 Myr in all simulations.

750 km ($\pm$ 80 km) in the horizontal direction, largely as a result of the varying SP dip/morphology arising from varying slab retreat rates. This far-field location is regarded as the edge of the flow cell in this work.

The Extension process is the competition of basal drag which thins the OP versus thermal healing which thickens it. The model goes to rift when the basal drag wins out, but thermal healing is always efficient because all the Extensions heal after a few Myrs as well. The thermal weakening starts at the base of the thermal boundary between the OP and the underlying mantle, though the viscosity weakens from both the base and the top of the OP (Figure 4) and produces 'necking' (Lei, 2022) in the middle, which is because yielding viscosity dominates at the top and it is not dependent on the temperature but on the yielding strain rate and the depth. Then the Extension appears from the base of the OP and the material flows into the gap at the surface. Thermal thinning gives thermal weakening, equally the hot region provides thermal thinning at the start of such simulations and encourages Extension. The detail of the effect of basal drag will depend upon the asthenosphere-lithosphere coupling, which is self-consistently solved for in our work which involves a thermal lithosphere but the basal drag could differ slightly in reality with more complex lithospheres.

From this discussion, we conclude that the primary extensional force probably arises from the basal drag, while the extension occurs when the OP is weakened.

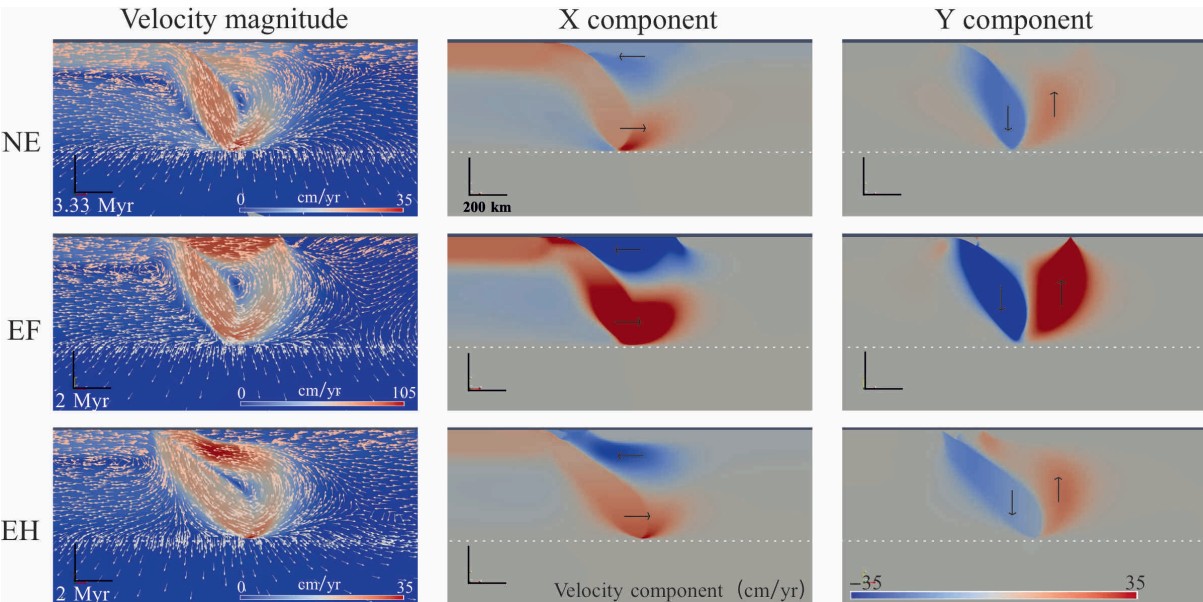

**Figure 13.** Velocity fields in 3 modes. The small arrows indicating the direction of the mantle flow are marked in the magnitude velocity field. In the velocity component field, the positive value represents rightwards (in the x component field) or upwards (in the y component field). The large arrows in the velocity component field show the velocity directions. The chosen time is just before the Extension (in Mode EF and EH) or Thinning (in Mode NE). Note that the scale of Velocity magnitude in Mode EF is set three times larger than that in other modes to improve figure clarity.

## 4.2 The role of an existing arc

In numerical models, the existence of an arc is not necessary to produce Extension in the OP (Figure 10; Sheng et al., 2019). However, the position of the Extension depends on the size of the mantle flow cell (about 750 km in our models) if the OP is homogeneous. A hot region on the OP not only facilitates the Extension to happen but also changes the position of the Extension under some conditions (Figure 6, 10). The specific mode that a model produces depends on different parameters, including properties of the hot region and plate ages in our cases. There have been many studies investigating back-arc extension (Sdrolias and Müller, 2006; Capitanio et al., 2010, 2011; Schellart and Moresi, 2013; Dal Zilio et al., 2018; Sheng et al., 2019; Dasgupta et al., 2021) and some have emphasised the balance of strength and forces that lead to extension and shown that the location of extension in the OP can be related to the flow cell in the mantle wedge (Dal Zilio et al., 2018). In our work here we similarly can find extension in a similar flow cell controlled location, but emplacement of a hot region can sufficiently weaken the OP to change the location of extension to this weakened region.

When the plate ages are fixed and the model shows Thinning but no Extension before a hot region is introduced, some models show Extension at the hot region (Mode EH), which means the hotter and weaker 'arc' is split apart; whereas some

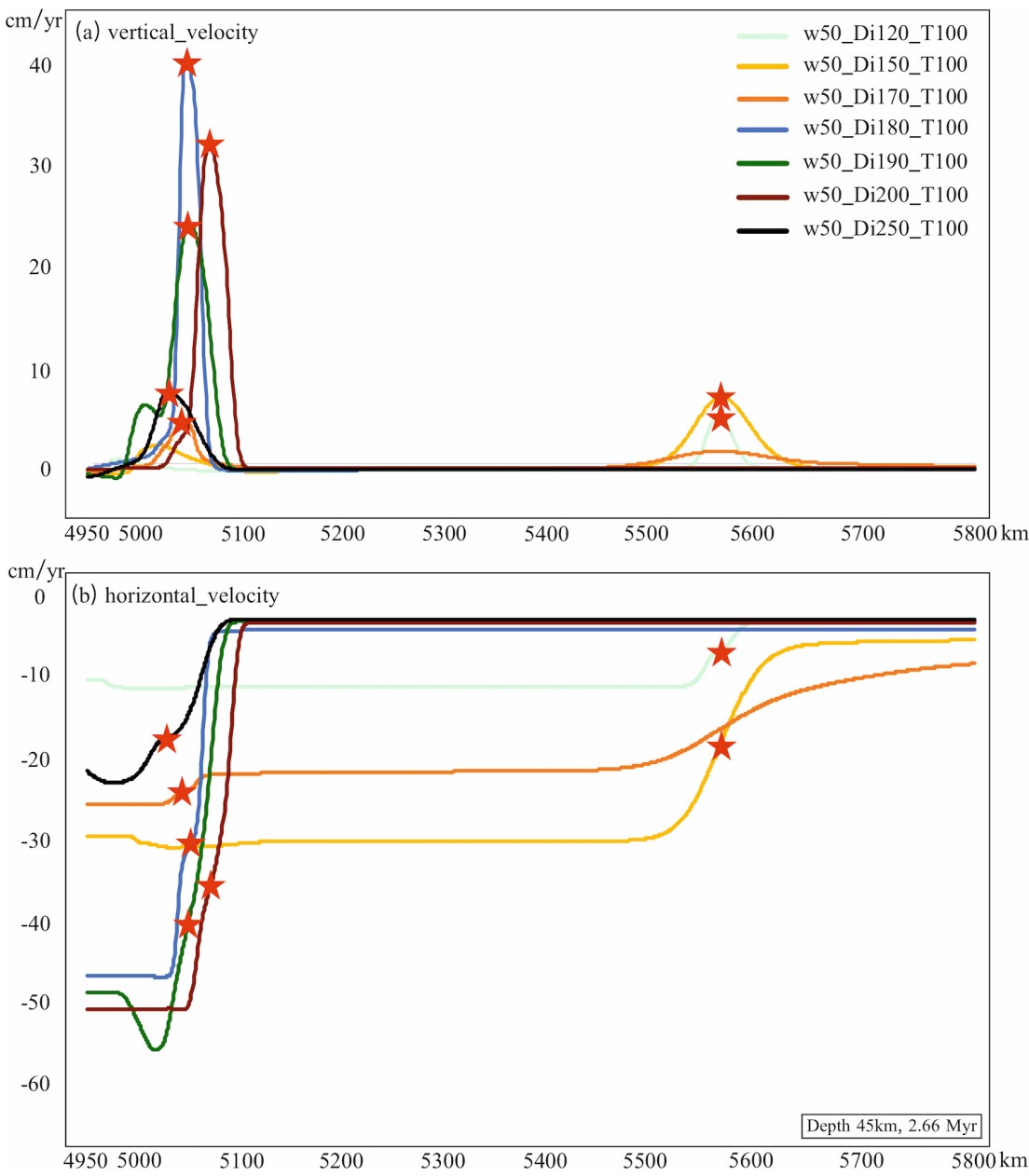

**Figure 14.** Velocity components vs. x coordinate at 45-km depth at 2.66 Myr. (a) Vertical velocity (Y component) vs. x coordinate; (b) Horizontal velocity (X component) vs. x coordinate. The back-arc extension is marked by red stars. Models W50_Di120_T100 and W50_Di150_T100 are in Mode EF, the others are all in Mode EH.

others show Extension at the same place as the thinning location even though the weakened zone is much closer to the trench (Mode EF), which implies the driving force (including the force which weakens the OP) is undoubtedly enhanced.

The Extension process is a competition of thermal weakening (reduced viscosity due to increased temperature) at the hot region and at the far-field location. The former is mainly from the thermal thinning of the hot region itself, so it depends on the Width and $\Delta T$ (the viscosity is dependent on temperature which is shown in Eq.6). The latter is from the mantle wedge flow, which is controlled by initial plate ages (which controls convergence rate) and Width and $\Delta T$ as well because faster trench retreat (Section 3.2.2) also induces a stronger mantle flow (Lei, 2022). The results show the contribution that Width and $\Delta T$
make to the thermal weakening of the hot region is greater than that to the wedge flow. The thinning is generated by thermal weakening. The thermal weakening draws in some flow because of the continuity of mass, which makes the hot region weaker and encourages more Extension. At the same time, the weakening accelerates the trench retreats and induces more wedge flow. When the mantle flow is firstly strengthened, the place which concentrates the flow at the far-field location is weakened and dragged by stronger basal traction no matter where the weak region is, which results in Mode EF. However, when the Width
and $\Delta T$ are high enough, the thermal weakening in the hot region dominates the process, and the hot region becomes the region of Extension, which is Mode EH (Figure 13). At the same time, the flow going to the far-field location reduces because the upwelling focuses on the hot region, so the far-field location does not show Extension anymore. If the thermal thinning which the mantle wedge flow produces exceeds that produced by the hot region, then Extension occurs at the far-field location.

    Different from the Width and $\Delta T$, the Distance influences neither the trench retreat rate nor the size of the mantle flow. The
320 upwelling flow is sucked into the edge of the flow cell and gradually decreases to each side (Figure 13). When the Distance increases, the hot region initially gets close to the cell edge and gains more upwelling intrusion. Hence, the hot region is increasingly weakened and attracts the mantle flow to itself. As the Distance increases further, the hot region moves away from the cell edge, and the mantle flow fades away, so Model W50_Di1050_T100 changes to EF mode again (Figure 7). However, a larger trench-arc distance is always associated with a flatter slab in nature (Cross and Pilger Jr, 1982), and there are positive
correlations between the slab dip and back-arc extension so that a BAB would be rarely observed when the arc is relatively far away from the trench.

    Figure 10 shows that when $\text{Age}_{SP}^0$ is relatively young (eg. 60-90 Ma), some cases which have an Extension at 750 km away from the trench with a homogeneous OP generate EH mode after adding the hot region. In these cases, a hot region largely weakens the OP and concentrates the flow at the weak zone though the arc also facilitates the mantle flow. When the arc is
330 much easier to be broken than the far-field location for this mantle flow, the flow tends to focus on the arc and the far-field location loses the extensional stress. The high negative buoyancy and strength of an older SP encourage a higher trench retreat rate and a stronger mantle flow (Garel et al., 2014), so that the flow is strong enough to break at the far-field location before the weak zone is broken. Under such circumstances, the models show EF mode. It is possible that at very old age the slab might resist bending leading to another mode (Di Giuseppe et al., 2009).

## 4.3   Comparison to observations in nature

Mode NE (No Extension), Mode EH (Extension at the Hot region) and Mode EF (Extension at Far-field location) are identified as three end-member modes (Section 3.1). In nature, similar behaviours to Mode NE and Mode EH are shown in many subduction zones. For example, the Lesser Antilles subduction zone does not show back-arc extension on the OP (Caribbean

Plate), which is consistent with Mode NE. The phenomenon that many BABs were formed by splitting a volcanic arc apart (Mariana Trough, Lau Basin, East Scotia Basin, etc) is reproduced by Mode EH. However, very few or even no BAB is definitely observed extending far away from the volcanic arc (like the Mode EF process). Speculating from the limited available information, the Japan Sea possibly formed far away behind the Japan Arc (Tatsumi et al., 1990) and might be an EF example. It is observed that the Japan Arc is approximately 300 km away from the Japan Trench (Jarrard, 1986), and the distance from the spreading centre of the Japan Sea to the trench is around 700 km (Tamaki, 1992), which is close to the mantle flow size in our models (750 km). Admittedly, there are opinions claiming that the Japan Sea opened at a proto-Japan Arc (Jolivet et al., 1994), but the evidence presented is debatable due to the lack of a remnant ridge on the western side of the Japan Sea.

Generally speaking, an old SP is more likely to generate back-arc extension in our models, which is consistent with observations in nature. For example, the East Pacific SPs are quite young at the interface where entering the subduction zones, corresponding to which there are few BABs found along the East Pacific margin. Specifically, the Juan de Fuca, the Explorer, the South Gorda and the Winona plates, which are along North America, are less than 10 Ma (Rogers, 1988), and the Nazca Plate which subducts beneath the South American Plate is about 40 Ma (Capitanio et al., 2011). In contrast, most western Pacific SPs are older than 100 Ma (Müller et al., 2008), and there are a lot of BABs distributed along that margin. This phenomenon meets our basic model behaviour. However, our models show Mode EF when the SP age is quite old because of the strong mantle flow it induces, whereas most of the BABs along the western Pacific formed in the process of Mode EH. To generate an EH, our models require an SP aged less than about 100 Ma and a young OP, or a hot region with a large size and high temperature when the SP is relatively old. In view of the ages of the western Pacific SPs (over 100 Ma), it is assumed that the arcs in nature would be hotter than that in our models. Unfortunately, the uncertainty in the effects of actual arcs on back-arc extensions makes it difficult to compare our model directly with observations.

Besides, the survival time of the extensions in our models is consistent with that in nature (Clark et al., 2008). Real back-arc basins are not active for a long time, mostly the Extension duration lasts less than 20 Myrs (Luyendyk et al., 1973; Tamaki, 1995; Fujioka et al., 1999; Jolivet and Faccenna, 2000; Eagles et al., 2005; Eagles, 2010; Barckhausen et al., 2014; Doo et al., 2015). For example, $^{40}$Ar-$^{39}$Ar dating indicated that the basement of the Japan Basin was aged 24-17 Ma, which means the crust grew for 7 Myrs (Kaneoka, 1992). The ages of the present-day BABs are usually less than 10 Myrs (Madsen and Lindley, 1994; Caratori Tontini et al., 2019), like the Mariana Trough started 5 Myrs ago (Yamazaki et al., 2003), the Okinawa Trough is only 2 Ma (Kimura, 1985; Sibuet et al., 1998), and the Lau Basin has been active for less than 6 Myrs (Weissel, 1977; Ruellan et al., 2003). The models of both EF and EH show that the Extension stops and tends to heal its thermal structure within 4-15 Myrs (older SP keeps the Extension longer), which is consistent with some cases in nature (eg. the Sulu Sea lasted for about 4 Myrs and Vavilov Basin lasted for 3-4 Myrs; Schliffke et al., 2022).

We must also acknowledge that the properties that a given arc would present to an overriding plate, in simulations like ours, are currently poorly defined. This further complicates, for now, a detailed comparison between models and nature.

## 4.4 Limitations

1. 2D modeling. (a) 2D models lack the lateral dimension of the real 3D world, which means the toroidal component of the mantle flow is ignored. There is some research suggesting that toroidal flow makes an important contribution to back-arc extension on the OP (Clio and Pieter, 2013; Chen et al., 2016), especially at narrow subduction zones (Schellart and Moresi, 2013). The hypothesis of our 2D models is that the width of plates is infinitely wide and everywhere is the same along the third dimension. Thus, the models may not be applicable to the narrow subduction system, such as Gibraltar and the Calabria subduction zones. (b) We are implicitly assuming that the arcs are linear and constant along strike, while in reality the arc volcanoes focus along strike. (c) The plates can be inhomogeneous perpendicular to strike, a direction which is modelled, but only with simply varying plates to allow comparison, or along strike, a direction that is not modelled.

2. Other simplifications of the models. (a) We did not run models with a real continental OP, even though the Japan Sea, a case with a continental OP, is discussed in this paper. Because the models are simplified to include only mantle (more viscous than crust at the same temperature) viscosity, the OP lithosphere is stronger than it should be at a given age (the thermal structure depends on the initial age of plates). A younger age reduces the OP thickness and decreases the total strength to partly counteract the lack of crust. As a result, the same phenomenon requires a real plate to be older than that in our models (and even older age for a continental plate compared to an oceanic plate). Similarly, we did not consider eclogitization, which would affect SP. (b) We have used a simple far-field boundary condition on the OP of a ridge, actual settings might be better represented by different boundary conditions (Capitanio et al., 2010; Nakakuki and Mura, 2013; Schellart and Moresi, 2013). We have kept the models simple in this case to allow comparison. (c) The process of forming an arc involves partial melting which can be expected to lower the viscosity and density in local regions (Corradino et al., 2022; Baitsch-Ghirardello et al., 2014). There is also a possibility for important feedback between the arc and subduction that we cannot capture in these models. (d) History of evolution is simple and similar to allow comparison across models, but this might be important in actual cases, For example, we miss out on the initiation stage of subduction and there is also a period of high subduction velocity in our models which might be higher than on Earth.

## 5 Conclusions

Our models show that a volcanic arc has a significant impact on the back-arc extension on the OP in a subduction system even though extension can also happen without the arc. It is worth noting that the conclusions are derived from 2D simulations with a mobile upper plate.

1. By means of introducing a hot region on the OP and testing various parameters, three end-member types of back-arc extension are depicted: (a) No back-arc Extension on the OP (Mode NE); (b) back-arc Extension occurs at the far-field location (about 750 km from the trench, Mode EF); and (c) back-arc Extension at the Hot region (Mode EH), including models with slab Detachment (EH-D). NE and EH modes are common in nature, whereas EF mode is very rare. We speculate that the Japan Sea might be a case of EF mode.

2. Properties of the hot region influence the modes. A farther distance from the hot region centre to the trench (Distance), a larger width of the hot region (Width), and a higher increased temperature of the hot region centre ($\Delta T$) encourage back-arc extension on the OP, as well as the modes transfer from (a) Mode NE, to (b) Mode EF, to (c) Mode EH.

3. Plate ages are important parameters influencing the mode changes as well. When the OP is homogeneous, an old SP or a young OP in a model encourages back-arc extension about 750 km from the trench on the OP (almost the same location as that in Mode EF). After introducing the hot region, we note that not only is back-arc extension more likely to happen, but the Extension switches from its far-field location into the hot region itself in some modes with a relatively young SP ($Age^0_{SP}$ less than 100 Ma in our models, but this threshold age would change if the hot region varies).

4. The primary driving mechanism of our models is poloidal flow underneath the OP. The flow cell has almost the same size in every model, which focuses at around 750 km from the trench on the OP and gradually decreases towards the trench. A higher trench retreat rate (through a higher slab sinking rate) facilitates the poloidal flow, thus encouraging back-arc extension.

Table A1: Models of SP90_OP20 with various hot region

| Models | Width (km) | Distance (km) | $\Delta T$ (°C) | Mode |
|---|---|---|---|---|
| w50_Di100_T50 | 50 | 100 | 50 | NE |
| w50_Di100_T800 | 50 | 100 | 100 | NE |
| w50_Di110_T200 | 50 | 110 | 200 | EF |
| w50_Di110_T300 | 50 | 110 | 300 | EF |
| w50_Di110_T800 | 50 | 110 | 800 | EF |
| w50_Di120_T50 | 50 | 120 | 50 | NE |
| w50_Di120_T100 | 50 | 120 | 100 | NE |
| w50_Di120_T200 | 50 | 120 | 200 | EF |
| w50_Di120_T800 | 50 | 120 | 800 | EF |
| w50_Di130_T50 | 50 | 130 | 50 | NE |
| w50_Di130_T100 | 50 | 130 | 100 | EF |
| w50_Di130_T200 | 50 | 130 | 200 | EF |
| w50_Di130_T300 | 50 | 130 | 300 | EF |
| w50_Di130_T400 | 50 | 130 | 400 | EF |
| w50_Di130_T500 | 50 | 130 | 500 | EH |
| w50_Di130_T600 | 50 | 130 | 600 | EH |
| w50_Di130_T700 | 50 | 130 | 700 | EH |
| w50_Di130_T800 | 50 | 130 | 800 | EH |
| w50_Di140_T50 | 50 | 140 | 50 | NE |
| w50_Di140_T100 | 50 | 140 | 100 | EF |
| w50_Di140_T300 | 50 | 140 | 300 | EH |

| | | | | |
|---|---|---|---|---|
| w50_Di150_T50 | 50 | 150 | 50 | EF |
| w50_Di150_T100 | 50 | 150 | 100 | EF |
| w50_Di150_T300 | 50 | 50 | 300 | EH |
| w50_Di160_T50 | 50 | 160 | 50 | EF |
| w50_Di160_T100 | 50 | 160 | 100 | EF |
| w50_Di160_T200 | 50 | 160 | 200 | EH |
| w50_Di160_T300 | 50 | 160 | 300 | EH |
| w50_Di170_T50 | 50 | 170 | 50 | EF |
| w50_Di170_T300 | 50 | 170 | 300 | EH |
| w50_Di180_T100 | 50 | 180 | 100 | EH |
| w50_Di180_T300 | 50 | 180 | 300 | EH |
| w50_Di190_T50 | 50 | 190 | 50 | EH |
| w50_Di190_T300 | 50 | 190 | 300 | EH |
| w50_Di200_T50 | 50 | 200 | 50 | EH |
| w50_Di200_T800 | 50 | 200 | 800 | EH |
| w50_Di400_T50 | 50 | 400 | 50 | EH |
| w50_Di750_T50 | 50 | 700 | 50 | EH |
| w50_Di750_T100 | 50 | 700 | 100 | EH |
| w50_Di750_T300 | 50 | 700 | 300 | EH |
| w50_Di800_T50 | 50 | 800 | 50 | EH |
| w50_Di800_T100 | 50 | 800 | 100 | EH |
| w50_Di800_T300 | 50 | 800 | 300 | EH |
| w50_Di900_T50 | 50 | 900 | 50 | EH |
| w50_Di900_T100 | 50 | 900 | 100 | EH |
| w50_Di900_T300 | 50 | 900 | 300 | EH |
| w50_Di1050_T50 | 50 | 1050 | 50 | EF |
| w50_Di1050_T100 | 50 | 1050 | 100 | EH |
| w50_Di1050_T300 | 50 | 1050 | 300 | EH |
| w25_Di125_T200 | 25 | 125 | 200 | NE |
| w25_Di125_T300 | 25 | 125 | 300 | EF |
| w25_Di150_T200 | 25 | 150 | 200 | EF |
| w25_Di150_T300 | 25 | 150 | 300 | EF |
| w25_Di150_T800 | 25 | 150 | 800 | EH |
| w25_Di175_T200 | 25 | 175 | 200 | EH |
| w25_Di175_T800 | 25 | 175 | 800 | EH |

| | | | | |
|---|---|---|---|---|
| w25_Di200_T800 | 25 | 200 | 800 | EH |

Table A2: Models testing plate ages with/without a hot region

| Models | SP age (Ma) | OP age (Ma) | hot region | Mode |
|---|---|---|---|---|
| SP55_OP15 | 55 | 15 | N | NE |
| SP60_OP15 | 60 | 15 | N | E |
| SP90_OP20 (RM) | 90 | 20 | N | NE |
| SP95_OP20 | 95 | 20 | N | E |
| SP150_OP25 | 150 | 25 | N | NE |
| SP160_OP25 | 160 | 25 | N | NE |
| SP170_OP25 | 170 | 25 | N | E |
| SP240_OP30 | 175 | 30 | N | NE |
| SP275_OP30 | 275 | 30 | N | NE |
| SP280_OP30 | 280 | 30 | N | E |
| SP300_OP30 | 300 | 30 | N | E |
| SP300_OP30 | 300 | 30 | N | E |
| hSP55_OP15 | 55 | 15 | Y | EH |
| hSP60_OP15 | 60 | 15 | Y | EH-D |
| hSP60_OP18 | 60 | 18 | Y | EH |
| hSP60_OP20 | 60 | 20 | Y | NE |
| hSP70_OP15 | 70 | 15 | Y | EH-D |
| hSP70_OP20 | 70 | 20 | Y | EH |
| hSP80_OP15 | 80 | 15 | Y | EH |
| hSP80_OP18 | 80 | 18 | Y | EH |
| hSP80_OP22 | 80 | 22 | Y | EH |
| hSP90_OP15 | 90 | 15 | Y | EH |
| hSP90_OP22 | 90 | 22 | Y | EH |
| hSP90_OP23 | 90 | 23 | Y | NE |
| hSP90_OP25 | 90 | 25 | Y | NE |
| hSP100_OP15 | 100 | 15 | Y | EF |
| hSP100_OP20 | 100 | 20 | Y | EH |
| hSP100_OP22 | 100 | 22 | Y | EF+EH |
| hSP100_OP23 | 100 | 23 | Y | NE |

| | | | | |
|---|---|---|---|---|
| hSP110_OP15 | 110 | 15 | Y | EF |
| hSP110_OP20 | 110 | 20 | Y | EH |
| hSP110_OP22 | 110 | 20 | Y | EF |
| hSP110_OP23 | 110 | 20 | Y | EF |
| hSP120_OP15 | 120 | 15 | Y | EF |
| hSP120_OP20 | 120 | 20 | Y | EF+EH |
| hSP120_OP22 | 120 | 22 | Y | EF |
| hSP120_OP23 | 120 | 23 | Y | EF |
| hSP120_OP25 | 120 | 25 | Y | NE |
| hSP130_OP20 | 130 | 20 | Y | EF |
| hSP150_OP15 | 150 | 15 | Y | EF+EH |
| hSP150_OP20 | 150 | 20 | Y | EF |
| hSP150_OP22 | 150 | 22 | Y | EF |
| hSP150_OP23 | 150 | 23 | Y | EF |
| hSP150_OP25 | 150 | 25 | Y | EF |
| hSP150_OP30 | 150 | 30 | Y | NE |
| hSP200_OP15 | 200 | 15 | Y | EF |
| hSP200_OP20 | 200 | 20 | Y | EF |
| hSP200_OP25 | 200 | 25 | Y | EF |
| hSP250_OP30 | 250 | 30 | Y | EF |
| hSP300_OP30 | 300 | 30 | Y | EF |
| hSP350_OP30 | 350 | 30 | Y | EF |

*Code and data availability.* The numerical simulations were run with the open-source and multiphase computational fluid dynamics code
Fluidity (http://fluidityproject.github.io/). The input files required to reproduce the simulations presented herein have also been made available: https://zenodo.org/records/11143014.

*Author contributions.* Duo Zhang: Writing the original draft & review & editing, Conceptualisation, Formal analysis, Investigation, Methodology, Visualisation, Validation, and Funding acquisition; Huw Davies: Writing - review & editing, Conceptualisation, Methodology, Validation, Supervision, Funding Acquisition, Project Administration, and Resources.

*Competing interests.* The contact author has declared that none of the authors has any competing interests.

*Acknowledgements.* This study is supported by PhD grant from the Chinese Scholarship Council (No. 201906170045), Cardiff University and supercomputer Hawk in Cardiff University. The authors would like to thank Attila Balazs, the anonymous reviewer(s) and the editor Taras Gerya for constructive comments that helped to improve the manuscript.

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
