# Peer review of "How a volcanic arc influences back-arc extension: insight from 2D numerical models"

_EGUsphere, 2023_

## Author Comment (AC1)

1. An advantage of this study in comparison with many other previous works is the application of an adaptive mesh that allows 400m spatial resolution along the subduction interface. It would be useful to include a bit more information about how the resolution changes over time and space. Does the domain of the finest resolution follow the changing location of the interface, for instance? The grid adapts throughout the simulation, keeping the finest resolution where the spatial gradients of fields are highest. The manuscript has been updated accordingly. So yes, the finest resolution follows the changing location of the interface.

2. Can you specify how the "thin weak layer" is built up to decouple the plates? Which rheology is used? The rheology of the subduction interface has a large impact on the stress transfer between the plates, also controlling back-arc deformation. Therefore, please discuss your subduction interface rheology, viscosity and compare it with previous studies. We decreased the maximum composite viscosity and the friction coefficient of yielding strength of the weak layer, and kept the diffusion, dislocation and Peierls creeps of the weak layer the same as that of the mantle material. We have edited the description of 'weak layer' in Section 2.1. (Lines 60-61)

3. The main method needs more justification: the location of the volcanic arc would be connected to the thermal regime of the overriding plate and the underlying mantle, as well as driven by the slab dip angle. How realistic is it to assume a fixed hot region for representing the volcanic arc? The hot-region is not fixed, but is just an initial thermal condition. A hot region is clearly an approximation to an arc, hence why we have investigated a range of initial conditions. Furthermore, why the 1300 K isotherm is not elevated in the "arc" region? The 1300 K isotherm is elevated in the "arc" region, but may not be that obvious. We have added zoom-in figures of the hot region in figure 4 (a) (c) (e) (g) (i). Does the chosen arc location fit the relationship of slab dip and arc location discussed by Ha et al. 2023 G3? Thank you for suggesting this paper. Yes, we have many cases which fit the geometry shown in Figure 9 of Ha et al. paper. We have run a series of models varying the arc-trench distance from 100 to 1000 km, which covers the range shown in this paper. Our slab dips vary from around 40 to 70 degrees, and are also similar to those in Ha et al. 2023. The authors should also reflect on previous works, where the volcanic arc formed self-consistently driven by the gradual hydration of the mantle wedge and modelling melt extraction. There are particular works, that addressed their role on upper plate rifting: e.g. Corradino et al. 2022 Sci. Rep.; Baitsch-Ghirardello et al. 2014 Gondwana Res. Thank you for letting us know about these works, we have added them to the text (lines 27-29 in the revised manuscript).

4. The authors use an ocean-ocean setup. Do the findings of the study applicable to a continental overriding plate? Thank you for this comment, this is a limitation of our research. As we mentioned in Limitation (Line 341-346 in original manuscript), we point out that a continental crust in the OP differs from our oceanic setup. Since crust is weaker than mantle and continental crust is thicker than oceanic crust, then we might expect continental lithosphere from a compositional perspective to be less viscous than oceanic lithosphere but thermally continental lithosphere can be older and hence colder and therefore could be expected to be more viscous than oceanic lithosphere. We would need to model this case to make definitive statements as to which is most important (composition or temperature), but it is reasonable to expect that the trends will be similar. Also since the OP weakens from its base, the nature of the near surface crust might be less critical. Therefore, while our findings might, to some extent, also be applicable to a continental overriding plate, we prefer to highlight this as a limitation.

5. I think the authors should better reflect on the limitation of using a 2D setup. The poloidal mantle flow is overpredicted in such 2D models, and there seems to be a broad consensus that in nature, back-arc extension is connected to the toroidal component of the mantle flow: e.g. McCabe 1984 Tectonics, followed by many modelling papers. This is connected to the 4th conclusions points, that back-arc extension is caused by the poloidal mantle flow. This is right, but rather a model limitation, thus I suggest moving it out from the conclusions. Thank you for this comment. Yes, toroidal flow plays an important role. The reason why we wrote about poloidal flow in the Conclusion is to note the importance of the size of the (poloidal) wedge flow cell and how this flow acts on the OP. It highlights the effect of poloidal flow. We think that the 2D setup is a significant limitation of this work, which we highlight in the Limitations sections.

6. As for the asthenosphere-lithosphere coupling: how would different mantle thermal gradients affect back-arc extension? Can you show a viscosity profile? Would a different profile, for instance, by assuming a different mantle thermal gradient or grain size evolution affect the coupling between the plate and underlying mantle? This should be mentioned at least in the discussion. Yes - it is possible that a different viscosity profile would change the coupling slightly, but you can see that there is a dramatic change in viscosity from the lithosphere to the asthenosphere, and therefore we do not expect changes that would change our conclusions. We present not just a profile but the whole viscosity field in figure 2b. The thermal structure though evolves according to the equations of physics and is not a free variable. We have added a brief comment in the discussion 4.1 – 'The detail of the effect of basal drag will depend upon the asthenosphere-lithosphere coupling, which is self-consistently solved for in our work which involves a thermal lithosphere but the basal drag could differ slightly in reality with more complex lithospheres.'

7. ln. 125: this means a subduction velocity larger than 11 cm/yr. Is it in agreement with observations and reconstructions? This is important, because the velocity of the induced poloidal flow would have similar values (in a 2D model) and this is linked to the potential coupling with the overriding plate. I suggest showing a plot on the relation between the modelled subduction velocity and upper plate lithosphere thinning over time. Yes, the peak subduction velocity in our models (it is worth noting that this lasts for only a short period of time - hence this period would be hard to identify on the seafloor of the related plate) is high compared to frequently quoted subduction velocities. The thinning occurs over a very short period of time, frequently close to the time of peak subduction velocity. For most of the early part of the simulations the plate thickness is constant or thickening very slightly by cooling. From animations (not presented, but happy to include in supplementary information if advised) of our simulations, it is clear that the OP thickness stays virtually constant and the subduction velocity increases as subduction progresses. Then in cases with extension the OP plate thins very quickly. Rather than present a plot of such simple behaviour we have described it in the text – 'We note that the SP sinking velocity in this short time period increases to a very high value (with local peak of > 10 cm/yr, and a lower peak when averaged peak over 1 Myr [geologicall observable timeframe], as expected, as the length of subducting slab increases, before decreasing to nearly steady values of < 2 cm/yr once the slab approaches and enters the more viscous lower mantle.' (Lines 133-136 in the revised manuscript.)

8. "Horizontal extensional force can be ignored as a cause of Extension in our models": this statement needs more attention, I suggest. Kinematically the retreating slab drives the divergence of the overriding plate. Extensional deformation will be localized along

the rheological weakest location. It is either along the imposed thermal weakness or the location overlying the mantle upwelling, connected to the return flow. Edited in Section 4.1 (Revised manuscript).

9. Instead of providing a list of a selected previous modelling papers (Line no. 15-17), it would be better and more useful to group them, which previous models contributed to which aspect of back-arc extension: e.g. analogue vs numerical, 2D vs 3D, assumed hydration and melting or not, used Newtonian or more complex rheologies, used spontaneous or forced subduction initiation, etc. Thank you for this suggestion. We have edited and grouped these papers by the method of simulation: analogue vs numerical models (Line 15-17 in the revised manuscript).

10. Subduction initiation would have an impact on the formation of an arc and also on the style of upper plate deformation (cf. Stern 2004, EPSL). In understand that this is not the primary topic of this manuscript. However, if one assumes spontaneous SI, by the time the leading edge of the slab reaches the prescribed 200 km, a back-arc spreading center could have been already formed. Thank you for this comment. Yes, this is a good point, but as you say it is not the primary topic of this manuscript. The work in this manuscript clearly would not be relevant to cases where a back arc spreading centre has already formed before the leading edge of the slab has reached 200 km. It is unclear whether this would be the case in spontaneous SI. Since our work does not address this case, we do not comment extensively, but accept it would be an excellent avenue of future research. We have added this text 'including for example missing out the initiation stage of subduction.' in the manuscript at lines 386-387 in the revised version, to remind the reader of this limitation.

11. The authors write that the overriding plate region in the close vicinity of the trench record compression before rifting. I don't think this is the artifact or due to the mentioned coarse time stepping. In our previous models (Balazs et al. 2022; Corradino et al. 2022) we visualized the stress field and the orientation of the principle stress axis and also found this compressional stress accumulation on the forearc region driven by vertical suction (resulting horizontal compression) of the slab. But, when the slab starts rolling back, of course, extensional deformation will be localized along the rheologically weakest part of the overriding plate, in your case, that is this region, where the "arc" was defined. Yes, we agree that the compression we have found near the trench before the extension is not due to the mentioned coarse time stepping and it is real and not an artifact. We note though that in Fig.10 in the original manuscript, the location where the EH happens recorded compression, but in fact we had incorrectly marked the extension location. It is marked correctly in Fig.11 (in revised manuscript) and is in a region under extension.

12. The statement in the introduction, that the nature of the overriding plate has not been extensively studied or the majority of the models listed above include a homogeneous OP is not the case. Just a few recent example: Wolf et al. 2019 JGR Solid Earth, Yang et al. 2021 G3. In our two papers on this topic: Balazs et al. 2022 Tectonics and Corradino et al. 2022 Sci. Rep., we particularly addressed the role of inherited structures, the formation of a volcanic arc and the possible locations of back-arc rifting. We have edited the text in Introduction (Lines 20-30 in the revised manuscript).

13. fig. 1: The location of the "arc" region is drawn above the slab in the zoomed image, while it is drawn as laterally shifted in the larger image. Thank you for pointing out this error. It has now been corrected.

14. no. 297-299: "The high negative buoyancy and strength of an older SP encourage a higher trench retreat rate and a stronger mantle flow (Garel et al., 2014), so that the flow is strong enough to break at the far-field location before the weak zone is broken. Under such circumstances, the models show EF mode." In fact, when the plate is too old and strong it resists to bend, therefore there is an optimum age, cf. Di Giuseppe et al. 2009 Lithosphere. Thank you for pointing it out. Yes, a plate will be hard to bend when its age is very old, although our tested ages of the SP are not old enough to get this phenomenon. Since the SP age on Earth is younger than our maximum tested SP age, we didn't mention this point. We have added to the text in lines 328-329 (revised version) mentioning this possibility - 'It is possible that at very old age the slab might resist bending leading to another mode.'

15. The text can be improved, for instance, this is not an optimal way of references: "and other references". The convention, the authors use to explain Complete Thinning and Spreading as "Extension" is misleading and not necessary. In the discussion, it is particularly challenging to follow the reasoning. I suggest simply using well established terms: when talking about strain: divergence or rifting, for processes spreading, post-rift relaxation, etc. Some sentences might be also simplified, like this: "The model goes to rift when the basal drag wins out, but thermal healing is always efficient because all models showing Extension show it healing after a few Myrs of Extension as well." We have decided to keep with our simple classification system of 2 states, Extension or No Extension. We note that Complete Thinning is Incipient Extension, and to make this clearer we have used the term Complete Thermal Thinning / Incipient Extension in Figure 3 and the text where we define these two states to make things clearer and hopefully less misleading.

16. To limitations: eclogitization? Partial melting: significantly drops viscosity and increases roll-back. Thank you, we have added these limitations in the 'Limitations' section. We have edited the text in Limitations as follows: "Similarly, we did not consider eclogitization, which would affect SP." "The process of forming an arc involves partial melting which can be expected to lower the viscosity and density in local regions. There is also a possibility for important feedback between the arc and subduction that we cannot capture in these models."

17. fig. 2: it is hardly possible to see the stress values in the overriding plate. This part should be enlarged and zoomed. Please indicate the horizontal and vertical scale in the figures. Thank you for the suggestion. We have added zoomed-in figures.

18. fig. 10: this figure should be placed in a supplementary material, and here you might rather show the models stress field and velocity field just before and after rifting. Yes, with the old figure 10, which was in error, we agree that a figure showing the model stress field and velocity field just before and after rifting would have helped show how this could be the case (such a result could have been correct - due to the possibility of non-uniform stress state with depth through the lithosphere) and was an excellent suggestion. Now that we have discovered the error in our original figure 10, (now figure 11) the updated correct figure provides a simple explanation of the stress state, and we prefer to keep it and not introduce more complex figures and text.

---

## Author Comment (AC2)

Second rewiewer:

The main ones are:

i) the explanation of the drivers of upper plate extension is hard to follow and I think, in parts, incorrect;

ii) there's a lack of justification/testing for the rheological properties of the upper plate (which are obviously important for whether a plate breaks or remains intact);

iii) the model velocities are extremely high but this is not discussed (but would affect upper plate extension significantly).

Main points:

Extension mechanism: You propose that extension can be triggered by either an extensional force in the plates, or subduction-induced flow beneath the plates, and rule out the first option (i.e., an extensional force, which you propose is related to the speed difference of the plates). This is misleading because extension will always be due to extensional normal stresses within the (pre-extended) plate. A more accurate way of framing this is whether this extensional normal stress is due to a horizontal force transmitted from the plate boundary (e.g., due to rollback) or, as you say, tractions from mantle flow. I don't disagree that basal tractions are the dominant control, just with how you are describing the different stress components/framing the physical problem. Also, about this mantle flow contribution, you state that it's dominantly the vertical (not horizontal) flow. But, prior to spreading, it's horizontal flow that produces basal tractions on the base of the near-flat lithosphere (which, yes, does ultimately originate from vertical flow that has been deflected horizontally). Yes, we agree that an extensional is always due to extensional normal stresses within the plate. Maybe we didn't describe it clearly but what we want to emphasise is that the extension is triggered by the decreasing strength of the OP but not increasing extensional force. We have edited the text in Discussion and hope it makes our point clearer.

There are a lot of modeling studies that delve into what dictates the upper plate stress state in dynamic subduction models (Capitanio et al., 2010, Tectonophys.; Schellart and Moresi, 2013, JGR; Holt et al., 2015, GJI; Dal Zilio et al., 2018, Tectonophys.) and carefully consider the relationship between sub-plate flow, basal tractions, and lithospheric stress. You cite some of these in passing; I recommend integrating the perspectives of these previous studies to present a clearer view of the forces in your upper plate and how they vary between models. An improved force description (Section 4.1), and an incorporation of this into Section 4.2, should make the discussion much clearer. We have edited our discussion. Also, in Figure 10, you plot integrated stress profiles and show that extension/spreading is triggered in a compressional region; this cannot be correct so should be sorted out (either by outputting more timesteps or plotting a zoom-in of the stress field for the corresponding timesteps to check your stress integration is correct). Sorry, Figure 10 in the original manuscript was in error, we wrongly marked the location of Extension in EH mode. Currently, the Extension is triggered in an extensional region, which is consistent with intuition.

Upper plate properties: You are investigating a balance between the forces driving the extension and the plate strength (i.e., extension when driving forces > strength) and so the

imposed strength of the upper plate is very important. I therefore recommend more discussion (or additional tests) of the parameters you choose that dictate this. It looks as though the non-extended (or pre-extended) lithospheric strength is dictated by the maximum imposed viscosity ($10^{25}$ Pa s) and that extension occurs once the stress > the plastic yield stress (2 MPa + 0.2 * Pressure). And while the hot zone will also lower the viscosity and so reduce the strength, you do not specify by how much. Given the importance, I recommend more discussion about your plastic yielding parameterization. What does this yielding viscosity represent? What dictated/justifies the parameter choices? You might run some tests to show that your first order findings do not depend on some of these choices too much. The plastic yielding parameterisation is very widely used, and can be thought to represent frictional failure. The choice of the 'friction coefficient', as mentioned in the text, falls between the estimates of Byerlee's law, and the much lower estimates of the friction coefficient at the SP/OP interface by other researchers. Since plate weakening dominates over increasing stresses, and plate weakening starts at the base of the OP, the critical rheologies are the viscous ones dominant at the base of the OP. Therefore, while the yielding will play a role in the total strength, and therefore how much stress is required to rift the plate, this will be secondary to the role of the other rheologies. We feel that if rheology parameters were to be investigated, they should all be investigated. That is beyond this work, but we agree is very worthy of investigation, and we have undertaken a separate project to this effect. We have also run some models increasing the maximum composite rheology and found that it makes no difference. The decrease in viscosity from the hot zone varies spatially according to the specific increase in temperature at that location, as given by the composite rheology. Therefore, it is not straightforward to specify by how much the viscosity is decreased - it is different within any case, and more so between cases. This is best done through figures and we have added the viscosity drop due to the hot zone in figure 4. In the caption of Figure 4: 'Figures (b) (d) (f) (h) (j) show the viscosity field and plots of the initial vertical viscosity at the hot region centre in the 0 Ma figure of each model represented by the red lines, while the black lines represent the viscosity without the hot region. The two lines in different colours show the change in viscosity resulting from emplacing a hot zone.'

Model velocities: You don't quote model convergence rates or trench motion rates but describe the slab hitting 660-km in 4 Myrs. This corresponds to very high sinking velocities (> 10 cm/yr) and very high mantle flow velocities (Figure 11). These velocities are likely very important in setting the stress in your models, and hence when extension/spreading occurs. I think should be discussed or, at the very least, explicitly pointed out. Yes, high sinking velocities (and therefore convergence velocities) do exist in these models for a short time as subduction starts, and drop significantly once the slab enters the lower mantle. Thank you for encouraging us to add some discussion on this point in the manuscript, added at lines 133-136. 'We note that the SP sinking velocity in this short time period increases to a very high value (with local peak of > 10 cm/yr, and a lower peak when averaged over 1 Myr [geologically observable timeframe]), as expected, as the length of subducting slab increases, before decreasing to nearly steady values of < 2 cm / yr once the slab approaches and enters the more viscous lower mantle.' This significant variation in sinking velocity is seen in many researchers work when they consider this early stage of subduction (Lei and Davies, 2023; Garel et al., 2014; Hall et al., 2003).

Line-by-line:

- L9: Is it a competition of "thermal weakening" between these two regions? Or is where has the largest extensional stress relative to the strength (which, at a certain location, is reduced due to the arc)? Yes, of course where extension occurs is strictly given by the statement included in the reviewer's second question. The extra weakening process though is the thermal weakening at both the hot-region and the far-field location. We have therefore decided to retain the statement since this is the main way that we describe the process in the article. We thank the reviewer for forcing us to consider this point.

- 22: Lots of studies looked at the controls on upper plate stress, so they did (albeit indirectly): e.g., those mentioned above. We have edited the text to now mention "many … previous models …". (Lines 20-30 in the revised manuscript.)

- 27-31: This passage summarizes the motivation/novelty very nicely (but I'm not sure what the Bettina reference is attached to). Dropped this citation off.

- ~47-49: Is it where the properties are changing the most quickly (as you write)? Or just spatial gradients at a given timestep? Yes, this is correct. Also, which properties do you use to refine? We refine at high spatial gradients of viscosity, second invariant of strain-rate, temperature and weak zone phase amount. We add the following sentence to address these points 'The grid adapts throughout the simulation, keeping the finest resolution where the spatial gradients of fields (viscosity, second invariant strain-rate, temperature and weak zone phase amount) are highest.' (Lines 54-55 in the revised manuscript.)

- 57: "around 194" -> "194". Edited.

- 66: What is this "prescribed depth"? The same as the depth of the plate bottom, which varies in different OP ages. We have edited in the text: 'The temperature at a chosen distance (from 100 to 1050 km) from the trench is increased by a certain degree (the degree varying from 25 to 800 degrees) vertically from the mantle depth (which varies in various OP ages) to the surface.' (Lines 70-71 in the revised manuscript.)

- Equation 4: You call $p$ both lithostatic and dynamic pressure. I think it's the "full" pressure (i.e., the sum of these two). We use the capital 'P' to represent the lithostatic pressure in Equation 9 and the small letter 'p' to represent the dynamic pressure in Equation 4.

- ~95: Where does this simplified parameterization of Peierls creep come from? Ref(s) needed. Added the description and reference in the text at Lines 116-117 in the revised manuscript.

- Equation 9: 2nd $\tau_y$ should be a $\tau_0$. Edited.

- 113: Viscous dashpots in series (i.e., the strain rates sum) not parallel. See Schmeling et al. (2008, PEPI). Thank you. The sentence has been edited to mention strain rate and be clearer. 'This is assuming that the strain rates of all 4 deformation processes sum, like viscous dashpots in series (Schmeling et al. 2008)'. (Lines 120-121 in the revised manuscript.)

- 160: about one-tenth -> one-tenth. Edited at Line 124 (revised version).

- 129: Difference between eroded and thinning? There is no fundamental difference between 'eroded' and 'thinning' in our work, so we have unified them to 'thinning' in the text (Line 137 and 140 in the revised version).

- Sect. 3.1: I think this description of the modes is quite confusing and can be simplified. Particularly the no extension vs. extension. E.g., on L139, you say that the state before complete thinning is classified as No Extension; but, on L141, you suggest that No Extension also corresponds to some thinning. I would just try and simplify this. Thank you for this comment. We have taken the opportunity to look at this again. We believe that having 2 states of extension is the simplest, leading to the 3 modes. We have re-read our description of our definition of these states at the first paragraph at the start of section 3.1 and feel that it is very clear. We state clearly that thinning is included as No Extension, while once it reaches complete thermal thinning and rifting it is Extension. We prefer to leave this classification but have changed 'Complete Thinning' to 'Complete Thermal Thinning and Incipient Extension' to make less confusion. We thank the reviewer for encouraging us to look at this again.

- Figure 2: I would add a length scale to the figures, particularly as you are talking about trench-extension distances. And I don't understand the stress units. We have added a length scale to all relevant figures and changed the stress unit to 'Pa'.

- 164-165: But is extension at the hot region (HR) always "close to the trench"? Because you are moving the HR quite far away, so it's quite far away? Find this confusing in your definition of EF vs. EH. No, HR can be far away from the trench. Our EF mode means the extension occurs both 'not at the HR' and 'far away from the trench'. There are some models in which the HR is far away from the trench and the back-arc extension occurs at the HR, these cases are called EH as well.

- 187: SP velocity, convergence rate, or slab sinking velocity? We thank the reviewer for pointing out the need for further clarification. This is in fact the vertical slab sinking velocity. The text is updated appropriately in Lines 199-200.

- 198-199: I think a weak OP just provides less horizontal resistance to rollback. E.g., single slabs models without OPs (e.g., old subduction models such as Enns et al. [2005, GJI]) always have high rollback as they are basically weak-OP endmembers. Thank you for the suggestion, we have edited the text in Line 211 (revised version).

- 230: The speed differences within the upper plate (as they produce horizontal normal strain rate) not the speed difference between the plates. We have edited the text in Lines 240-242: 'In the first mechanism the extensional stress is generated from the speed difference between the trench and the OP, while the second arises from variable shear stresses at the base of the OP.'

- 237-239 and Figure 10: It's hard to see what the issue is – Can you also show zoomed-in plots of the horizontal stress field at equivalent times, e.g., as the Schellart & Moresi paper does. Perhaps you are outputting the stress after the extension has occurred? Instead of right before. Sorry, we mislocated one star marker in Fig.10 in the original manuscript. We have edited it in the figure and updated the relevant text (Line 253-259 in revised manuscript).

- 247: As mentioned, I think the upwelling flow would weaken the upper plate via basal drag (i.e., these two things are one mechanism). Unless you are talking about after extension, and during spreading, when the upwelling would sustain spreading. But it's not really clear from the text explanation. Yes, we do not mean to ignore the importance of the basal drag but wanted to emphasise the thermal weakening caused by the upward flow. We have edited the text at Line 267-269 by adding a comment to make clear that the basal drag can also weaken the OP, and emphasise that both components ultimately result from upwelling flow. 'The trench-ward horizontal flow

(X component) produces basal drag by the velocity gradient (Figure 13), which facilitates an Extension by producing the extensional force and weakening the OP, whereas the Y component of the flow also encourages the thermal weakening of the OP. The upwelling flow, the cause of both components of flow, is always in a similar direction.'

- Figure 11: is the velocity scale really up to 105 cm/yr?! Or is this a typo? No, it is not a typo, it is up to this value, but only very briefly. We have discussed these high velocity values earlier in the response and added a comment in the manuscript.

- 265-266: I think incorporating the results of these studies (and Capitanio et al., 2010; Schellart and Moresi, 2013, etc.) would make this discussion a bit clearer from a mechanism point of view. Those studies outline where (and how much) extension we get in OPs; you are then effectively combining this logic (about driving forces) with a weak zone of various magnitudes. Thank you. We have changed the text to take up your suggestion and added your suggested reference. 'There have been many studies investigating back-arc extension and some have emphasised the balance of strength and forces that lead to extension and shown that the location of extension in the OP can be related to the flow cell in the mantle wedge. In our work here we similarly can find extension in the same flow cell controlled location, but emplacement of a hot region can sufficiently weaken the OP to change the location of extension to this weakened region.'

- 271: What is meant by "thermal weakening"? We have added a description in the text 'reduced viscosity due to increased temperature'. (Line 300 in revised manuscript)

- 274: In 2-D, the mantle wedge flow will be more strongly controlled by the convergence or slab sinking rate. See the classic corner flow papers by McKenzie, Tovish and Schubert, etc. Yes, agreed. We have edited the text to reinforce the importance of convergence rate (Line 303 in revised manuscript).

- 286: Upwelling "intrusion" is maybe confusing when talking about the drivers of extension – the intrusion only happens after extension has occurred and spreading has created the space. Edited 'intrusion' to 'flow'.

- 323-329: What are the references for these back-arc basin ages? Sdrolias and Muller? Also see Clarke, Stegman, Muller (2008, PEPI). We have added some references to these back-arc basin ages (Lines 354-363 in revised manuscript).

- Section 4.4: In this limitation sections, I recommend: i) avoiding the double list (i.e., two layers of numbering); ii) including refences to studies that have considered these complexities; iii) organizing it more intuitively (e.g., going from simplifications that you think are the most important to those that are the least). We have preferred to keep the double list (but changing one set of numbers to letters) to group the limitations. Within this constraint we have tried to go from the most important to the least important. We have added a few references that have considered these complexities (Lines 381-384 in revised manuscript).

- 369: A higher slab sinking/convergence rate (which likely coincides with a high trench retreat rate). Thank you. We have clarified this by adding 'through a higher slab sinking rate' to the text (Line 406 in revised manuscript).

- 370: What does you will share your models "upon reasonable request" mean? Consider sharing your input files in an open online repository. If anybody contacts me, the

corresponding author, then I will share with them any model outputs they are interested in.

---

## Author Response (AR3)

**First Revision**

**Reviewer #1 (Attila Balazs):**

1. An advantage of this study in comparison with many other previous works is the application of an adaptive mesh that allows 400m spatial resolution along the subduction interface. It would be useful to include a bit more information about how the resolution changes over time and space. Does the domain of the finest resolution follow the changing location of the interface, for instance? The grid adapts throughout the simulation, keeping the finest resolution where the spatial gradients of fields are highest. The manuscript has been updated accordingly. So yes, the finest resolution follows the changing location of the interface.

2. Can you specify how the "thin weak layer" is built up to decouple the plates? Which rheology is used? The rheology of the subduction interface has a large impact on the stress transfer between the plates, also controlling back-arc deformation. Therefore, please discuss your subduction interface rheology, viscosity and compare it with previous studies. We decreased the maximum composite viscosity and the friction coefficient of yielding strength of the weak layer, and kept the diffusion, dislocation and Peierls creeps of the weak layer the same as that of the mantle material. We have edited the description of 'weak layer' in Section 2.1. (Lines 60-61)

3. The main method needs more justification: the location of the volcanic arc would be connected to the thermal regime of the overriding plate and the underlying mantle, as well as driven by the slab dip angle. How realistic is it to assume a fixed hot region for representing the volcanic arc? The hot-region is not fixed, but is just an initial thermal condition. A hot region is clearly an approximation to an arc, hence why we have investigated a range of initial conditions. Furthermore, why the 1300 K isotherm is not elevated in the "arc" region? The 1300 K isotherm is elevated in the "arc" region, but may not be that obvious. We have added zoom-in figures of the hot region in figure 4 (a) (c) (e) (g) (i). Does the chosen arc location fit the relationship of slab dip and arc location discussed by Ha et al. 2023 G3? Thank you for suggesting this paper. Yes, we have many cases which fit the geometry shown in Figure 9 of Ha et al. paper. We have run a series of models varying the arc-trench distance from 100 to 1000 km, which covers the range shown in this paper. Our slab dips vary from around 40 to 70 degrees, and are also similar to those in Ha et al. 2023. The authors should also reflect on previous works, where the volcanic arc formed self-consistently driven by the gradual hydration of the mantle wedge and modelling melt extraction. There are particular works, that addressed their role on upper plate rifting: e.g. Corradino et al. 2022 Sci. Rep.; Baitsch-Ghirardello et al. 2014 Gondwana Res. Thank you for letting us know about these works, we have added them to the text (lines 27-29 in the revised manuscript).

4. The authors use an ocean-ocean setup. Do the findings of the study applicable to a continental overriding plate? Thank you for this comment, this is a limitation of our research. As we mentioned in Limitation (Line 341-346 in original manuscript), we point out that a continental crust in the OP differs from our oceanic setup. Since crust is weaker than mantle and continental crust is thicker than oceanic crust, then we might expect continental lithosphere from a compositional perspective to be less viscous than oceanic lithosphere but thermally continental lithosphere can be older and hence colder and therefore could be expected to be more viscous than oceanic lithosphere. We would need to model this case to make definitive statements as to which is most important

(composition or temperature), but it is reasonable to expect that the trends will be similar. Also since the OP weakens from its base, the nature of the near surface crust might be less critical. Therefore, while our findings might, to some extent, also be applicable to a continental overriding plate, we prefer to highlight this as a limitation.

5. I think the authors should better reflect on the limitation of using a 2D setup. The poloidal mantle flow is overpredicted in such 2D models, and there seems to be a broad consensus that in nature, back-arc extension is connected to the toroidal component of the mantle flow: e.g. McCabe 1984 Tectonics, followed by many modelling papers. This is connected to the 4th conclusions points, that back-arc extension is caused by the poloidal mantle flow. This is right, but rather a model limitation, thus I suggest moving it out from the conclusions. Thank you for this comment. Yes, toroidal flow plays an important role. The reason why we wrote about poloidal flow in the Conclusion is to note the importance of the size of the (poloidal) wedge flow cell and how this flow acts on the OP. It highlights the effect of poloidal flow. We think that the 2D setup is a significant limitation of this work, which we highlight in the Limitations sections.

6. As for the asthenosphere-lithosphere coupling: how would different mantle thermal gradients affect back-arc extension? Can you show a viscosity profile? Would a different profile, for instance, by assuming a different mantle thermal gradient or grain size evolution affect the coupling between the plate and underlying mantle? This should be mentioned at least in the discussion. Yes - it is possible that a different viscosity profile would change the coupling slightly, but you can see that there is a dramatic change in viscosity from the lithosphere to the asthenosphere, and therefore we do not expect changes that would change our conclusions. We present not just a profile but the whole viscosity field in figure 2b. The thermal structure though evolves according to the equations of physics and is not a free variable. We have added a brief comment in the discussion 4.1 – 'The detail of the effect of basal drag will depend upon the asthenosphere-lithosphere coupling, which is self-consistently solved for in our work which involves a thermal lithosphere but the basal drag could differ slightly in reality with more complex lithospheres.'

7. ln. 125: this means a subduction velocity larger than 11 cm/yr. Is it in agreement with observations and reconstructions? This is important, because the velocity of the induced poloidal flow would have similar values (in a 2D model) and this is linked to the potential coupling with the overriding plate. I suggest showing a plot on the relation between the modelled subduction velocity and upper plate lithosphere thinning over time. Yes, the peak subduction velocity in our models (it is worth noting that this lasts for only a short period of time - hence this period would be hard to identify on the seafloor of the related plate) is high compared to frequently quoted subduction velocities. The thinning occurs over a very short period of time, frequently close to the time of peak subduction velocity. For most of the early part of the simulations the plate thickness is constant or thickening very slightly by cooling. From animations (not presented, but happy to include in supplementary information if advised) of our simulations, it is clear that the OP thickness stays virtually constant and the subduction velocity increases as subduction progresses. Then in cases with extension the OP plate thins very quickly. Rather than present a plot of such simple behaviour we have described it in the text – 'We note that the SP sinking velocity in this short time period increases to a very high value (with local peak of > 10 cm/yr, and a lower peak when averaged peak over 1 Myr [geologicall observable timeframe], as expected, as the length of subducting slab increases, before decreasing to nearly steady values of < 2

cm/yr once the slab approaches and enters the more viscous lower mantle.' (Lines 133-136 in the revised manuscript.)

8. "Horizontal extensional force can be ignored as a cause of Extension in our models": this statement needs more attention, I suggest. Kinematically the retreating slab drives the divergence of the overriding plate. Extensional deformation will be localized along the rheological weakest location. It is either along the imposed thermal weakness or the location overlying the mantle upwelling, connected to the return flow. Edited in Section 4.1 (Revised manuscript).

9. Instead of providing a list of a selected previous modelling papers (Line no. 15-17), it would be better and more useful to group them, which previous models contributed to which aspect of back-arc extension: e.g. analogue vs numerical, 2D vs 3D, assumed hydration and melting or not, used Newtonian or more complex rheologies, used spontaneous or forced subduction initiation, etc. Thank you for this suggestion. We have edited and grouped these papers by the method of simulation: analogue vs numerical models (Line 15-17 in the revised manuscript).

10. Subduction initiation would have an impact on the formation of an arc and also on the style of upper plate deformation (cf. Stern 2004, EPSL). In understand that this is not the primary topic of this manuscript. However, if one assumes spontaneous SI, by the time the leading edge of the slab reaches the prescribed 200 km, a back-arc spreading center could have been already formed. Thank you for this comment. Yes, this is a good point, but as you say it is not the primary topic of this manuscript. The work in this manuscript clearly would not be relevant to cases where a back arc spreading centre has already formed before the leading edge of the slab has reached 200 km. It is unclear whether this would be the case in spontaneous SI. Since our work does not address this case, we do not comment extensively, but accept it would be an excellent avenue of future research. We have added this text 'including for example missing out the initiation stage of subduction.' in the manuscript at lines 386-387 in the revised version, to remind the reader of this limitation.

11. The authors write that the overriding plate region in the close vicinity of the trench record compression before rifting. I don't think this is the artifact or due to the mentioned coarse time stepping. In our previous models (Balazs et al. 2022; Corradino et al. 2022) we visualized the stress field and the orientation of the principle stress axis and also found this compressional stress accumulation on the forearc region driven by vertical suction (resulting horizontal compression) of the slab. But, when the slab starts rolling back, of course, extensional deformation will be localized along the rheologically weakest part of the overriding plate, in your case, that is this region, where the "arc" was defined. Yes, we agree that the compression we have found near the trench before the extension is not due to the mentioned coarse time stepping and it is real and not an artifact. We note though that in Fig.10 in the original manuscript, the location where the EH happens recorded compression, but in fact we had incorrectly marked the extension location. It is marked correctly in Fig.11 (in revised manuscript) and is in a region under extension.

12. The statement in the introduction, that the nature of the overriding plate has not been extensively studied or the majority of the models listed above include a homogeneous OP is not the case. Just a few recent example: Wolf et al. 2019 JGR Solid Earth, Yang et al. 2021 G3. In our two papers on this topic: Balazs et al. 2022 Tectonics and Corradino et al. 2022 Sci. Rep., we particularly addressed the role of inherited structures,

the formation of a volcanic arc and the possible locations of back-arc rifting. We have edited the text in Introduction (Lines 20-30 in the revised manuscript).

13. fig. 1: The location of the "arc" region is drawn above the slab in the zoomed image, while it is drawn as laterally shifted in the larger image. Thank you for pointing out this error. It has now been corrected.

14. no. 297-299: "The high negative buoyancy and strength of an older SP encourage a higher trench retreat rate and a stronger mantle flow (Garel et al., 2014), so that the flow is strong enough to break at the far-field location before the weak zone is broken. Under such circumstances, the models show EF mode." In fact, when the plate is too old and strong it resists to bend, therefore there is an optimum age, cf. Di Giuseppe et al. 2009 Lithosphere. Thank you for pointing it out. Yes, a plate will be hard to bend when its age is very old, although our tested ages of the SP are not old enough to get this phenomenon. Since the SP age on Earth is younger than our maximum tested SP age, we didn't mention this point. We have added to the text in lines 328-329 (revised version) mentioning this possibility - 'It is possible that at very old age the slab might resist bending leading to another mode.'

15. The text can be improved, for instance, this is not an optimal way of references: "and other references". The convention, the authors use to explain Complete Thinning and Spreading as "Extension" is misleading and not necessary. In the discussion, it is particularly challenging to follow the reasoning. I suggest simply using well established terms: when talking about strain: divergence or rifting, for processes spreading, post-rift relaxation, etc. Some sentences might be also simplified, like this: "The model goes to rift when the basal drag wins out, but thermal healing is always efficient because all models showing Extension show it healing after a few Myrs of Extension as well." We have decided to keep with our simple classification system of 2 states, Extension or No Extension. We note that Complete Thinning is Incipient Extension, and to make this clearer we have used the term Complete Thermal Thinning / Incipient Extension in Figure 3 and the text where we define these two states to make things clearer and hopefully less misleading.

16. To limitations: eclogitization? Partial melting: significantly drops viscosity and increases roll-back. Thank you, we have added these limitations in the 'Limitations' section. We have edited the text in Limitations as follows: "Similarly, we did not consider eclogitization, which would affect SP." "The process of forming an arc involves partial melting which can be expected to lower the viscosity and density in local regions. There is also a possibility for important feedback between the arc and subduction that we cannot capture in these models."

17. fig. 2: it is hardly possible to see the stress values in the overriding plate. This part should be enlarged and zoomed. Please indicate the horizontal and vertical scale in the figures. Thank you for the suggestion. We have added zoomed-in figures.

18. fig. 10: this figure should be placed in a supplementary material, and here you might rather show the models stress field and velocity field just before and after rifting. Yes, with the old figure 10, which was in error, we agree that a figure showing the model stress field and velocity field just before and after rifting would have helped show how this could be the case (such a result could have been correct - due to the possibility of non-uniform stress state with depth through the lithosphere) and was an excellent suggestion. Now that we have discovered the error in our original figure 10, (now figure

11) the updated correct figure provides a simple explanation of the stress state, and we prefer to keep it and not introduce more complex figures and text.

**Reviewer #2:**

The main ones are:

i) the explanation of the drivers of upper plate extension is hard to follow and I think, in parts, incorrect;

ii) there's a lack of justification/testing for the rheological properties of the upper plate (which are obviously important for whether a plate breaks or remains intact);

iii) the model velocities are extremely high but this is not discussed (but would affect upper plate extension significantly).

Main points:

Extension mechanism: You propose that extension can be triggered by either an extensional force in the plates, or subduction-induced flow beneath the plates, and rule out the first option (i.e., an extensional force, which you propose is related to the speed difference of the plates). This is misleading because extension will always be due to extensional normal stresses within the (pre-extended) plate. A more accurate way of framing this is whether this extensional normal stress is due to a horizontal force transmitted from the plate boundary (e.g., due to rollback) or, as you say, tractions from mantle flow. I don't disagree that basal tractions are the dominant control, just with how you are describing the different stress components/framing the physical problem. Also, about this mantle flow contribution, you state that it's dominantly the vertical (not horizontal) flow. But, prior to spreading, it's horizontal flow that produces basal tractions on the base of the near-flat lithosphere (which, yes, does ultimately originate from vertical flow that has been deflected horizontally). Yes, we agree that an extensional is always due to extensional normal stresses within the plate. Maybe we didn't describe it clearly but what we want to emphasise is that the extension is triggered by the decreasing strength of the OP but not increasing extensional force. We have edited the text in Discussion and hope it makes our point clearer.

There are a lot of modeling studies that delve into what dictates the upper plate stress state in dynamic subduction models (Capitanio et al., 2010, Tectonophys.; Schellart and Moresi, 2013, JGR; Holt et al., 2015, GJI; Dal Zilio et al., 2018, Tectonophys.) and carefully consider the relationship between sub-plate flow, basal tractions, and lithospheric stress. You cite some of these in passing; I recommend integrating the perspectives of these previous studies to present a clearer view of the forces in your upper plate and how they vary between models. An improved force description (Section 4.1), and an incorporation of this into Section 4.2, should make the discussion much clearer. We have edited our discussion. Also, in Figure 10, you plot integrated stress profiles and show that extension/spreading is triggered in a compressional region; this cannot be correct so should be sorted out (either by outputting more timesteps or plotting a zoom-in of the stress field for the corresponding timesteps to check your stress integration is correct). Sorry, Figure 10 in the original manuscript was in error, we wrongly

marked the location of Extension in EH mode. Currently, the Extension is triggered in an extensional region, which is consistent with intuition.

Upper plate properties: You are investigating a balance between the forces driving the extension and the plate strength (i.e., extension when driving forces > strength) and so the imposed strength of the upper plate is very important. I therefore recommend more discussion (or additional tests) of the parameters you choose that dictate this. It looks as though the non-extended (or pre-extended) lithospheric strength is dictated by the maximum imposed viscosity ($10^{25}$ Pa s) and that extension occurs once the stress > the plastic yield stress (2 MPa + 0.2 * Pressure). And while the hot zone will also lower the viscosity and so reduce the strength, you do not specify by how much. Given the importance, I recommend more discussion about your plastic yielding parameterization. What does this yielding viscosity represent? What dictated/justifies the parameter choices? You might run some tests to show that your first order findings do not depend on some of these choices too much. The plastic yielding parameterisation is very widely used, and can be thought to represent frictional failure. The choice of the 'friction coefficient', as mentioned in the text, falls between the estimates of Byerlee's law, and the much lower estimates of the friction coefficient at the SP/OP interface by other researchers. Since plate weakening dominates over increasing stresses, and plate weakening starts at the base of the OP, the critical rheologies are the viscous ones dominant at the base of the OP. Therefore, while the yielding will play a role in the total strength, and therefore how much stress is required to rift the plate, this will be secondary to the role of the other rheologies. We feel that if rheology parameters were to be investigated, they should all be investigated. That is beyond this work, but we agree is very worthy of investigation, and we have undertaken a separate project to this effect. We have also run some models increasing the maximum composite rheology and found that it makes no difference. The decrease in viscosity from the hot zone varies spatially according to the specific increase in temperature at that location, as given by the composite rheology. Therefore, it is not straightforward to specify by how much the viscosity is decreased - it is different within any case, and more so between cases. This is best done through figures and we have added the viscosity drop due to the hot zone in figure 4. In the caption of Figure 4: 'Figures (b) (d) (f) (h) (j) show the viscosity field and plots of the initial vertical viscosity at the hot region centre in the 0 Ma figure of each model represented by the red lines, while the black lines represent the viscosity without the hot region. The two lines in different colours show the change in viscosity resulting from emplacing a hot zone.'

Model velocities: You don't quote model convergence rates or trench motion rates but describe the slab hitting 660-km in 4 Myrs. This corresponds to very high sinking velocities (> 10 cm/yr) and very high mantle flow velocities (Figure 11). These velocities are likely very important in setting the stress in your models, and hence when extension/spreading occurs. I think should be discussed or, at the very least, explicitly pointed out. Yes, high sinking velocities (and therefore convergence velocities) do exist in these models for a short time as subduction starts, and drop significantly once the slab enters the lower mantle. Thank you for encouraging us to add some discussion on this point in the manuscript, added at lines 133-136. 'We note that the SP sinking velocity in this short time period increases to a very high value (with local peak of > 10 cm/yr, and a lower peak when averaged over 1 Myr [geologically observable timeframe]), as expected, as the length of subducting slab increases, before decreasing to nearly steady values of < 2 cm / yr once the slab approaches and enters the more viscous lower mantle.' This significant variation in sinking velocity is seen in many researchers work when they consider this early stage of subduction (Lei and Davies, 2023; Garel et al., 2014; Hall et al., 2003).

Line-by-line:

- L9: Is it a competition of "thermal weakening" between these two regions? Or is where has the largest extensional stress relative to the strength (which, at a certain location, is reduced due to the arc)? Yes, of course where extension occurs is strictly given by the statement included in the reviewer's second question. The extra weakening process though is the thermal weakening at both the hot-region and the far-field location. We have therefore decided to retain the statement since this is the main way that we describe the process in the article. We thank the reviewer for forcing us to consider this point.

- 22: Lots of studies looked at the controls on upper plate stress, so they did (albeit indirectly): e.g., those mentioned above. We have edited the text to now mention "many … previous models …". (Lines 20-30 in the revised manuscript.)

- 27-31: This passage summarizes the motivation/novelty very nicely (but I'm not sure what the Bettina reference is attached to). Dropped this citation off.

- ~47-49: Is it where the properties are changing the most quickly (as you write)? Or just spatial gradients at a given timestep? Yes, this is correct. Also, which properties do you use to refine? We refine at high spatial gradients of viscosity, second invariant of strain-rate, temperature and weak zone phase amount. We add the following sentence to address these points 'The grid adapts throughout the simulation, keeping the finest resolution where the spatial gradients of fields (viscosity, second invariant strain-rate, temperature and weak zone phase amount) are highest.' (Lines 54-55 in the revised manuscript.)

- 57: "around 194" -> "194". Edited.

- 66: What is this "prescribed depth"? The same as the depth of the plate bottom, which varies in different OP ages. We have edited in the text: 'The temperature at a chosen distance (from 100 to 1050 km) from the trench is increased by a certain degree (the degree varying from 25 to 800 degrees) vertically from the mantle depth (which varies in various OP ages) to the surface.' (Lines 70-71 in the revised manuscript.)

- Equation 4: You call p both lithostatic and dynamic pressure. I think it's the "full" pressure (i.e., the sum of these two). We use the capital 'P' to represent the lithostatic pressure in Equation 9 and the small letter 'p' to represent the dynamic pressure in Equation 4.

- ~95: Where does this simplified parameterization of Peierls creep come from? Ref(s) needed. Added the description and reference in the text at Lines 116-117 in the revised manuscript.

- Equation 9: $2^{nd}$ tau_y should be a tau_0. Edited.

- 113: Viscous dashpots in series (i.e., the strain rates sum) not parallel. See Schmeling et al. (2008, PEPI). Thank you. The sentence has been edited to mention strain rate and be clearer. 'This is assuming that the strain rates of all 4 deformation processes sum, like viscous dashpots in series (Schmeling et al. 2008)'. (Lines 120-121 in the revised manuscript.)

- 160: about one-tenth -> one-tenth. Edited at Line 124 (revised version).

- 129: Difference between eroded and thinning? There is no fundamental difference between 'eroded' and 'thinning' in our work, so we have unified them to 'thinning' in the text (Line 137 and 140 in the revised version).

- Sect. 3.1: I think this description of the modes is quite confusing and can be simplified. Particularly the no extension vs. extension. E.g., on L139, you say that the state before complete thinning is classified as No Extension; but, on L141, you suggest that No Extension also corresponds to some thinning. I would just try and simplify this. Thank you for this comment. We have taken the opportunity to look at this again. We believe that having 2 states of extension is the simplest, leading to the 3 modes. We have re-read our description of our definition of these states at the first paragraph at the start of section 3.1 and feel that it is very clear. We state clearly that thinning is included as No Extension, while once it reaches complete thermal thinning and rifting it is Extension. We prefer to leave this classification but have changed 'Complete Thinning' to 'Complete Thermal Thinning and Incipient Extension' to make less confusion. We thank the reviewer for encouraging us to look at this again.

- Figure 2: I would add a length scale to the figures, particularly as you are talking about trench-extension distances. And I don't understand the stress units. We have added a length scale to all relevant figures and changed the stress unit to 'Pa'.

- 164-165: But is extension at the hot region (HR) always "close to the trench"? Because you are moving the HR quite far away, so it's quite far away? Find this confusing in your definition of EF vs. EH. No, HR can be far away from the trench. Our EF mode means the extension occurs both 'not at the HR' and 'far away from the trench'. There are some models in which the HR is far away from the trench and the back-arc extension occurs at the HR, these cases are called EH as well.

- 187: SP velocity, convergence rate, or slab sinking velocity? We thank the reviewer for pointing out the need for further clarification. This is in fact the vertical slab sinking velocity. The text is updated appropriately in Lines 199-200.

- 198-199: I think a weak OP just provides less horizonal resistance to rollback. E.g., single slabs models without OPs (e.g., old subduction models such as Enns et al. [2005, GJI]) always have high rollback as they are basically weak-OP endmembers. Thank you for the suggestion, we have edited the text in Line 211 (revised version).

- 230: The speed differences within the upper plate (as they produce horizontal normal strain rate) not the speed difference between the plates. We have edited the text in Lines 240-242: 'In the first mechanism the extensional stress is generated from the speed difference between the trench and the OP, while the second arises from variable shear stresses at the base of the OP.'

- 237-239 and Figure 10: It's hard to see what the issue is – Can you also show zoomed-in plots of the horizontal stress field at equivalent times, e.g., as the Schellart & Moresi paper does. Perhaps you are outputting the stress after the extension has occurred? Instead of right before. Sorry, we mislocated one star marker in Fig.10 in the original manuscript. We have edited it in the figure and updated the relevant text (Line 253-259 in revised manuscript).

- 247: As mentioned, I think the upwelling flow would weaken the upper plate via basal drag (i.e., these two things are one mechanism). Unless you are talking about after extension, and during spreading, when the upwelling would sustain spreading. But it's not really clear from the text explanation. Yes, we do not mean to ignore the importance of the basal drag but wanted to emphasise the thermal weakening caused

by the upward flow. We have edited the text at Line 267-269 by adding a comment to make clear that the basal drag can also weaken the OP, and emphasise that both components ultimately result from upwelling flow. 'The trench-ward horizontal flow (X component) produces basal drag by the velocity gradient (Figure 13), which facilitates an Extension by producing the extensional force and weakening the OP, whereas the Y component of the flow also encourages the thermal weakening of the OP. The upwelling flow, the cause of both components of flow, is always in a similar direction.'

- Figure 11: is the velocity scale really up to 105 cm/yr?! Or is this a typo? No, it is not a typo, it is up to this value, but only very briefly. We have discussed these high velocity values earlier in the response and added a comment in the manuscript.

- 265-266: I think incorporating the results of these studies (and Capitanio et al., 2010; Schellart and Moresi, 2013, etc.) would make this discussion a bit clearer from a mechanism point of view. Those studies outline where (and how much) extension we get in OPs; you are then effectively combining this logic (about driving forces) with a weak zone of various magnitudes. Thank you. We have changed the text to take up your suggestion and added your suggested reference. 'There have been many studies investigating back-arc extension and some have emphasised the balance of strength and forces that lead to extension and shown that the location of extension in the OP can be related to the flow cell in the mantle wedge. In our work here we similarly can find extension in the same flow cell controlled location, but emplacement of a hot region can sufficiently weaken the OP to change the location of extension to this weakened region.'

- 271: What is meant by "thermal weakening"? We have added a description in the text 'reduced viscosity due to increased temperature'. (Line 300 in revised manuscript)

- 274: In 2-D, the mantle wedge flow will be more strongly controlled by the convergence or slab sinking rate. See the classic corner flow papers by McKenzie, Tovish and Schubert, etc. Yes, agreed. We have edited the text to reinforce the importance of convergence rate (Line 303 in revised manuscript).

- 286: Upwelling "intrusion" is maybe confusing when talking about the drivers of extension – the intrusion only happens after extension has occurred and spreading has created the space. Edited 'intrusion' to 'flow'.

- 323-329: What are the references for these back-arc basin ages? Sdrolias and Muller? Also see Clarke, Stegman, Muller (2008, PEPI). We have added some references to these back-arc basin ages (Lines 354-363 in revised manuscript).

- Section 4.4: In this limitation sections, I recommend: i) avoiding the double list (i.e., two layers of numbering); ii) including refences to studies that have considered these complexities; iii) organizing it more intuitively (e.g., going from simplifications that you think are the most important to those that are the least). We have preferred to keep the double list (but changing one set of numbers to letters) to group the limitations. Within this constraint we have tried to go from the most important to the least important. We have added a few references that have considered these complexities (Lines 381-384 in revised manuscript).

- 369: A higher slab sinking/convergence rate (which likely coincides with a high trench retreat rate). Thank you. We have clarified this by adding 'through a higher slab sinking rate' to the text (Line 406 in revised manuscript).

- 370: What does you will share your models "upon reasonable request" mean? Consider sharing your input files in an open online repository. If anybody contacts me, the corresponding author, then I will share with them any model outputs they are interested in.

**Second Revision**

**Reviewer #1 (Attila Balazs):**

The authors well responded to my comments and the revised manuscript reads well. This paper provides important insights into the driving mechanisms behind back-arc extension. Before accepting the manuscript, I would only suggest the following minor comments:

1. the importance of the manuscript is about analyzing the stress field evolution. However, the simulated stress field is only in a too small figure. I zoomed to 400% to Fig. 2 and still struggled to see the values of the stress field as written in the text. Please either make a separate figure of this or really zoom into the lithospheric domain that is discussed. For instance at 12 Myr, does the horizontal stress field shows contrasting extensional vs compressional stress on the two sides of the thinned area and in the upper vs lower part of the lithosphere? Can you comment this? Authors have edited the stress field (zoomed in further and used a different colour bar to make the stress clearer). The region that undergoes thinning is no longer thinning but healing at 4 Myr, and the stresses are not driving any more deformation after that, so the horizontal stress then does not need to be the peak extensional horizontal stress. We didn't show the time point just before the thinning, because the main purpose of this figure is to show the overall evolution of the subduction zone and not the full history of the stress field (we note that the history of this stress is summarised in Figure 11).

2. Fig. 4: there are plots of the viscosities based on the figure caption, but those are too small and not clear. Thank you for the suggestion. The authors have reduced the numbers of the snapshots and split figure 4 into 2 figures to make everything easier to read.

3. Please add scale and coordinates in fig. 10c. Edited in what is now fig.11.

Conclusions: Please also state here that your conclusions are derived from 2D simulations with a mobile upper plate. Edited in the text (Line 396-397).

**Reviewer #2:**

I previously reviewed the original submission (2nd reviewer) and am now reviewing the original submission. I'm sorry for the delay with both of my reviews! It's been a really busy period. My main concerns centered around the mechanism descriptions, upper plate rheology, and the very rapid subducting plate velocities. The authors have worked on addressing each of these concerns but, in my opinion, a bit more work is needed for this to be sufficient. I'd therefore recommend "moderate" revisions. Below are my line-by-line comments:

Before the line-by-line: In code availability, you state that "the results of our models are available from the corresponding author upon reasonable request". I recommend sharing the full input files in a permanent citable repository, as is now standard practice (and I think the

Solid Earth data policy requires this). See recent geodynamics Fluidity modeling studies that do this (e.g., Chen et al., 2022, https://doi.org/10.1029/2022GC010757). OK, thank you, the authors will do this.

Abstract: Confusing to write "Mode EH" and "Mode EF" in the abstract - just describe. Authors have edited these references in the text (Line 3-5 in the revised version).

Line 48: How is it "based on the work of Garel et al." Do you mean that you started with their models and then added the hot regions? I recommend being more precise. Yes, that's what the authors mean. In section 2.1, the authors have described it by introducing Garel's model firstly and then the hot region. Thank you for your suggestion which gives us a chance to have a look at this text again. For a more precise description, the authors have edited Section 2.1 (Line 60-61 in the revised version) and hope it would be clearer to readers.

Eq. 4 and surrounding text: The pressure description is still confused. The "full" stress is made of the full pressure (lithostatic + dynamic) and the deviatoric stress. Thank you for pointing out the potential confusion. '$\sigma_{ij}$' is the stress tensor, where the lithostatic pressure is implicitly removed since only the lateral density variations are applied on the right-hand side of equation 2. So Eq.4 should use dynamic pressure. We have edited lines 90-91 in the revised version to make the description of the pressure and stress clearer.

L100-105: Where are these parameters taken from? Dry or wet olivine? Needs a reference. These parameters are from experimental estimations of dry olivine (Hirth and Kohlstedt, 2003; Karato and Wu, 1993; Karato, 1997). Authors have added these references in the text (Line 105-106 in the revised version).

Eq. 9 and surrounding text: One of my previous comments related to justifying the yield stresses imposed in the models. I understood this is a common approach – and that it's intended to mimic brittle failure – but I was more interested in how you justify these parameter values (cohesion, friction coefficient). Can you provide further details / reference(s) here? Many papers don't provide such details but, in the case of your study, it's especially important as the focus is on breaking the plate. Otherwise, you could show tests showing that the first-order conclusions do not depend on mu = 0.2 vs., e.g., mu = 0.4 (as you state in your original rebuttal). The friction coefficient of 0.2 in our models is intermediate between lower values of previous subduction models (Di Giuseppe et al., 2008; Crameri et al., 2012). Authors have added this note and references into the text (Line 120).

L120: "like viscous dashpots in series" -> "as is the case for viscous dashpots in series". Authors have edited it in the text (Line 125).

L132-136: As also identified by both reviewers, the subduction velocities are extremely large: this is likely to promote excessive extension (as stresses scale approx. with velocities). Given this, I think more details are needed: You say that there is a local peak of "> 10 cm/yr". What is the local peak, actually? Thank you for this question. The actual peak velocity varies between models, here it is about 38 cm/yr in the reference model. And does extension always coincide with this peak? These details should be added to the main text. Yes, we note that in cases with extension this coincides closely with this peak (this is shown in fig.9 in the revised version), they both happen just before the slab tip reaches the more viscous lower mantle. We didn't emphasise this point because the extension doesn't always appear at the same time as

the velocity peaks, but around it (a little before or after), so we think they cannot be strictly related, but we have qualified the text on line 141 to point to this broad correlation in time.

Figure 2: Stress should be in MPa (i.e., +/- 350 MPa). There is no velocity vector scale on the bottom panels. And what is "velocity component" (velocity magnitude?) We have changed the stress scale to MPa. Sorry, this figure does not show a velocity component, authors have edited 'velocity component' to 'velocity magnitude' in the figure. And why is it limited to 3.5 cm/yr (when you've stated that velocities are much higher)? In the reference model, the peak velocity is not that high. The high velocities always appear in models with weaker overriding plate and older (more negativey buoyant) subducting slabs. Also we should note, since the very high peak velocities are only very brief features, that this relatively higher velocit is between the outputs we have shown and so we didn't catch that the peak velocity.

Figure 4: The new panels (viscosity profiles) are way too small to read. The viscosity panels are probably also too small. Break the figure up into 2? Thank you for the suggestion. Author have reduced the numbers of the snapshots and split figure 4 into 2 figures to make everything easier to read.

L198: Can you quote some trench retreat velocity magnitudes? That's so readers can compare with other models/observations. Thank you for this suggestion. Authors have quoted the trench retreat rate in bracket. (Line 203)

L241: Saying that, in the first mechanism, the extensional stress is generated from the speed difference between the trench and the OP is not strictly true. That would thicken/thin the plate interface, which is not what you are talking about. We prefer to keep the current wording, noting that the 'trench' could be considered to be on the OP side of the plate interface. We are looking at the largest scale processes acting on the OP, and believe in this context this description is correct.

243: "a little more" – too informal. Authors have edited the whole sentence to 'We further investigate these aspects to find which are most important in our models.' (Line 248)

Figure 10: Panel B looks bad – can you plot in the same format as panel A. Also, did you integrate over the same depth range? Or the depth of the plate (which varies with age)? If the latter, I recommend dividing by this plate thickness to get the average plate stress. We have altered panel B of old figure 10 / new figure 11 to be in the same format as panel A. We integrated the stress over the depth of the plate which varies with age. Thank you for your suggestion and we have edited the figure to display the average plate stress.

253-255: I don't think this is correct (similar point to my original review). Yes, this shows that an in-plate stress increase is not the main extension initiation; rather it's the weakening in the upper plate. Agreed. But it does not discriminate between extension driven by force transmitted through the interface and that driven by underlying tractions. Both induce horizontal extension in the OP. Thanks for pointing it out. Authors have deleted 'applied at the OP/SP interface' in Line 264 and hoped that would reduce confusion.

354: Clarke, Stegman, Muller (2008, PEPI) is a nice paper about this. Their work should be referenced. We added this reference into the text. (Line 359)

Sect. 4.4 (Limitations): Bullet points 1 ("2D modeling") and 2 ("Simplications of the models") overlap (as 2-D is also a modeling simplification). More importantly, and following previous comments, I think one of the biggest simplifications is the combination of a very high subduction velocity and a relatively low yield stress (which both promote extension). I think this should be acknowledged. Thank you for these suggestions. To solve the overlapped classification, we have added 'other' before 'simplification of the models' (Line 380). We also have added the high subduction velocity into the limitation (Line 392-393), but we do not consider the maximum yield stress of 10 GPa as being particularly low (10 GPa) nor the friction coefficient.